# Test-time Training for Matching-based Video Object Segmentation

**Juliette Bertrand**[*1,2]    **Giorgos Kordopatis-Zilos**[*1]    **Yannis Kalantidis**[2]    **Giorgos Tolias**[1]

[1]VRG, FEE, Czech Technical University in Prague    [2]NAVER LABS Europe

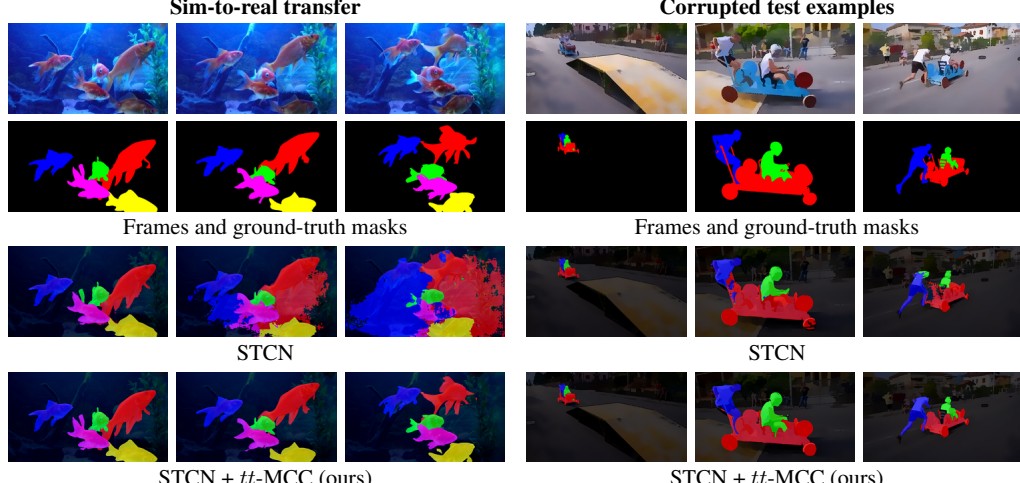

Figure 1: **Test-time training** significantly improves performance in the presence of distribution shifts for the task of video object segmentation (VOS). *Left:* Example using a model trained on synthetic videos from BL-30K [8] and tested on a real video from DAVIS [38]. *Right:* Example using a model trained on DAVIS [38] and YouTube-VOS [51] and tested on a cartoonized video from the **corrupted DAVIS-C benchmark** which we introduce in this work. Second-to-bottom row: Results obtained using the STCN [9] approach. Bottom row: Results after test-time training with the proposed *mask cycle consistency* loss ($tt$-MCC) using the single ground-truth mask provided for the first video frame.

## Abstract

The video object segmentation (VOS) task involves the segmentation of an object over time based on a single initial mask. Current state-of-the-art approaches use a memory of previously processed frames and rely on matching to estimate segmentation masks of subsequent frames. Lacking any adaptation mechanism, such methods are prone to test-time distribution shifts. This work focuses on matching-based VOS under distribution shifts such as video corruptions, stylization, and sim-to-real transfer. We explore test-time training strategies that are agnostic to the specific task as well as strategies that are designed specifically for VOS. This includes a variant based on mask cycle consistency tailored to matching-based VOS methods. The experimental results on common benchmarks demonstrate that the proposed test-time training yields significant improvements in performance. In particular for the sim-to-real scenario and despite using only a single test video, our approach manages to recover a substantial portion of the performance gain achieved through training on real videos. Additionally, we introduce DAVIS-C, an augmented version of the popular DAVIS test set, featuring extreme distribution shifts like image-/video-level corruptions and stylizations. Our results illustrate that test-time training enhances performance even in these challenging cases. Project page: https://jbertrand89.github.io/test-time-training-vos/

37th Conference on Neural Information Processing Systems (NeurIPS 2023).

---

*  Equal contribution

# 1 Introduction

In one-shot video object segmentation (VOS), we are provided with a single segmentation mask for one or more objects in the first frame of a video, which we need to segment across all video frames. It is a dense prediction task over time and space, and therefore, collecting training data is highly demanding. Early VOS methods design foreground/background segmentation models that operate on single-frame input and require a two-stage training process [4, 29]. During the off-line stage, the model is trained to segment a variety of foreground objects. During the on-line stage for a specific test video, the model is fine-tuned to adapt to the objects of interest indicated in the provided mask.

Recently, *matching-based* methods like STCN [9] or XMem [6] have shown impressive performance on the common VOS benchmarks. These methods propagate the segmentation mask from a memory of previously predicted masks to the current frame. They only involve an off-line training stage where the model learns how to perform matching and propagation. Lacking any test-time adaptation mechanism, however, such methods are highly prone to test-time distribution shifts.

Our goal is to improve the generalization of matching-based VOS methods under distribution shifts by fine-tuning the model at test time through a single test video. Such one-shot adaptation is a form of *test-time training* (TTT), a research direction that is lately attracting much attention in classification tasks and a promising way for generalizing to previously unseen distributions. We argue that TTT is a great fit for matching-based VOS, not only because of the additional temporal dimension to exploit, but also because we are given a ground-truth label for a frame of the test example.

In this work, we focus on two cases of test-time distribution shift for VOS: a) when training on synthetic data and testing on real data, also known as *sim-to-real transfer*, and b) when the test data contain image/video corruption or stylization. We explore three different ways of performing TTT: i) a task-agnostic approach that borrows from the image classification literature and performs entropy minimization [34, 46], ii) an approach tailored to the STCN architecture that employs an auto-encoder loss on the mask provided for the first frame and does not exploit any temporal information, and iii) a mask cycle consistency approach that is tailored to matching-based VOS, and utilizes temporal cycle consistency of segmentation masks to update the model. As we show, these different test-time training strategies do not work equally well, and tailoring TTT to the task and the methods is important and not trivial.

Our experiments demonstrate that some of the proposed approaches constitute a powerful way of generalizing to unseen distributions. When starting from a model trained only on synthetic videos, test-time training based on mask cycle consistency improves STCN by +22 and +11 points in terms of $\mathcal{J}\&\mathcal{F}$ score on the two most popular VOS benchmarks, YouTube-VOS and DAVIS, respectively. What is even more impressive is that by simply using test-time training, we recover the bulk of the performance gain that training the original model on real videos brings: for YouTube-VOS and DAVIS, TTT recovers 82% and 72% of the performance gains, respectively, that training on real videos brings compared to synthetic ones, but *without requiring any video annotations during off-line training*.

We also study the performance of TTT in the presence of image-level or video-level corruptions and stylization. To that end, we follow a process similar to ImageNet-C [14] and create DAVIS-C, a version of the DAVIS test set with 14 corruptions and style changes at three strength levels. We evaluate models with and without TTT on DAVIS-C and analyze how performance changes as the distribution shift increases. Our results show that TTT significantly improves the performance of STCN by more than 8 points in terms of $\mathcal{J}\&\mathcal{F}$ score for the hardest case of corruption/stylization. A qualitative example for both types of distribution shift is shown in Figure 1.

# 2 Related Work

**Foreground/background models with online fine-tuning for VOS.** Early approaches train a segmentation model to separate the foreground object from the background. The offline training stage is followed by an online supervised fine-tuning stage that uses the mask of the first frame for adapting the model [4, 29]. The fine-tuning process is improved by including pseudo-masks derived from highly confident predictions [44] or by additionally including hard-negative examples [48]. Maninis et al. [33] additionally incorporate semantic information about the underlined object categories via an external instance-aware semantic segmentation model. Other methods first produce a coarse segmentation result and then refine it by the guidance of optical flow [15], the appearance cue only [39], or temporal

consistency [32]. Hu et al. [16] utilize an RNN model to exploit the long-term temporal structures, while others additionally incorporate optical flow information [16]. The heavy cost of fine-tuning is reduced by meta-learning in the work of Xiao et al. [49] or Meinhardt and Leal-Taixé [35].

**Matching-based VOS.** Most recent approaches build upon a matching-based prediction model, which is trained offline and neither requires nor incorporates any fine-tuning stage. The current test frame is matched to either the first annotated frame [52, 18, 5, 45, 3], a propagated mask [20, 17, 28], or a memory containing both [31, 36, 9]. Early methods use simple matching process [18] or are inspired by classical label propagation [60], while different methods improve the design of the matching [1, 9, 40, 53, 55, 54], the type and extent of the memory used [6], or exploit interactions between objects or frames [37, 56], among other aspects.

**Hybrid and other VOS methods.** Mao et al. [34] borrow from both families of approaches, *i.e.* matching-based and those that require online fine-tuning, and jointly integrate two models, but , unlike our work, the matching model is not updated at test time. Some methods differ from matching-based in the way they capture the evolution over time, *e.g.* by RNNs [43] or by spatio-temporal object-level attention [11].

**Cycle consistency in VOS.** Matching-based methods require dense video annotations for training, which is a limitation. To dispense with the need for mask annotations during training, cycle consistency of pixel or region correspondences over space and time has been successfully used by prior work [19, 24, 61]. This is an unsupervised loss that is also appropriate for test-time training. In our case, we tailor cycle consistency to the task of matching-based VOS, and propose a *supervised* consistency loss that operates on masks by taking advantage of the provided mask at test-time in order to better adapt to the object of interest. Mask consistency based on the first frame is used by prior work during model training. Khoreva et al. [21] synthesize future frames using appearance and motion priors, while in our case, instead of synthesizing, we use mask predictions of the existing future video frames. Li et al. [26] and HODOR [1] use a temporal cycle consistency loss on a the first frame mask during training that allows learning from videos with a single segmentation mask. In contrast, we utilize the mask frame that is provided at test-time for the first frame of the test video and we are able to outperform HODOR and other state-of-the-art methods for multiple test-time distribution shifts.

**Test-time training.** A family of approaches adapts the test example to the training domain. Kim et al. [22] chose the appropriate image augmentation that maximizes the loss according to a loss prediction function, while Xiao et al. [50] updates the features of the test sample by energy minimization with stochastic gradient Langevin dynamics. Another family of approaches adapts the model to the test domain. Entropy minimization is a common way to update either batch-normalization statistics only [46] or the whole model [59]. Other self-supervised objectives include rotation prediction [42], contrastive learning [30], or spatial auto-encoding [12]. In our work, we move beyond the image domain and introduce mask cycle consistency as an objective to adapt the model specifically for video object segmentation applications. Azimi et al. [2] evaluate some of the aforementioned TTT techniques on top of self-supervised dense tracking methods under several distribution shifts on videos. Another recent work [57], concurrent to ours, also studies TTT in the video domain but for a classification task; we instead fully tailor the TTT on the VOS task and use temporal *mask* consistency as our loss.

## 3 Test-time training for matching-based VOS

In this section, we first formulate the task of Video Object Segmentation (VOS) and briefly describe the basic components of the matching-based VOS methods we build on [9, 6]. We then present three ways for test-time training: a task-agnostic baseline based on entropy minimization that has been highly successful in other tasks and two VOS-specific variants that leverage the fact that we are provided with a ground-truth mask at test time; one using an auto-encoder loss and another a temporal cycle-consistency loss on the masks.

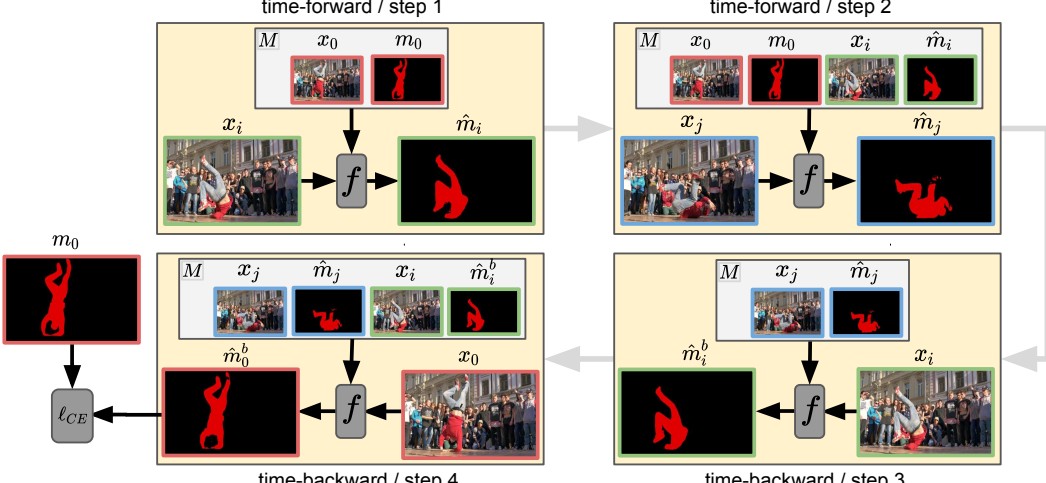

Figure 2: **The mask cycle consistency loss** for a given frame triplet $\{x_0, x_i, x_j\}$ and mask $m_0$. Frames $x_0, x_i$ and $x_j$ are shown with a red, green and blue border, respectively. Function $f$ represents the overall prediction model. It takes as input the current test frame and a memory $M$ of predicted masks from the previous frames and outputs the predicted mask for the current frame. For the first frame, the memory contains the ground-truth mask.

## 3.1 Preliminaries

The goal of VOS is to segment a set of particular objects in a video provided their ground-truth segmentation mask $m_0$ in the first video frame $x_0$[1]. Space-Time Correspondence Networks (STCN) [9] sequentially predict the mask of each frame separately, while the prediction is conditioned on the provided mask and the predicted masks of earlier frames that are stored in a memory queue. XMem [6] is an extension of STCN that features a long-term memory and top performance for long-term prediction. Although we provide here the necessary parts of STCN required for presenting our contribution, we refer the reader to Cheng et al. [9] for a complete description. Our formulation is generic and applicable to a wide range of matching-based VOS methods, including the ones that require only a single forward pass for multiple objects such as AOT [56].

We denote the model that performs the overall process by function $f$. The predicted mask of frame $x_j$ is given by $\hat{m}_j = f(x_j; M)$, where $M = \{(x_i, m_i)\}$ is the memory of previous predictions consisting of frame-mask tuples $(x_i, m_i)$. The memory always includes $(x_0, m_0)$, while any other tuple for $i > 0$ is formed by using the corresponding predicted mask $\hat{m}_i$ since no other ground-truth mask is available. Note that ground-truth masks for the first frame are binary, while the predicted ones consist of object probabilities in $[0, 1]$.

The STCN model contains a frame encoder $e_x$ that extracts frame representation $u_i = e_x(x_i) \in \mathbb{R}^{W \times H \times D_e}$ and a mask encoder $e_m$ that extracts mask representation $v_i = e_m(m_i) \in \mathbb{R}^{W \times H \times D_m}$. The latter can take either ground-truth (binary) or predicted (real-value) masks as input. Frame and mask representations of earlier frames are added to the memory.

During the prediction for the current frame denoted by $x_j$, representation $u_j$ is matched (each $D_e$-dimensional vector separately) to the representations of frames in the memory, and the most similar items are identified. In particular, the full similarity matrix is formed between the current frame items and the memory items, but only the top-k values per row (rows correspond to the current frame) are maintained, and the rest are set to 0. By the term item we refer to each $D_e$-dimensional vector, with each frame having $W * H$ of them. The corresponding mask representation of these most similar items is aggregated to obtain $\hat{v}_j \in \mathbb{R}^{W \times H \times D_m}$. In this way we compose the mask representation for frame $x_j$ via matching since the mask itself is not available. This process is

---

[1]In the following, we assume the presence of a single object for simplicity. In the case of multiple objects, the prediction process is repeated for each object independently, similar to [6, 9].

equivalent to a cross-attention operation between the test frame features (queries) and the memory frame features (keys) that aggregates memory mask features (values). Finally, a mask decoder $d_m$ is used to obtain the predicted mask $\hat{m}_j = d_m([\hat{v}_j, u_j])$, where $[\hat{v}_j, u_j] \in \mathbb{R}^{W \times H \times (D_e + D_m)}$ is the concatenation of the two representations along the depth dimension.

In this work, we assume that we have access to a matching-based VOS model, trained on real or synthetic data with mask annotations. These models do not generalize well to extreme distribution shifts not encountered during supervised training. Our goal is to improve their performance in such cases by fine-tuning the parameters of the model at test time.

## 3.2 Test-time training

We explore three different losses to perform TTT for the case of matching-based VOS: (i) based on entropy minimization (*tt-Ent*), which forms a paradigm transferred from the image domain (image classification in particular) and is a task- and method-agnostic loss, (ii) based on an auto-encoder (*tt-AE*) that operates on segmentation masks, which is an approach tailored for the STCN architecture, (iii) based on mask cycle consistency (*tt-MCC*), which is more general and appropriate for a variety of matching-based methods. *tt*-MCC exploits the matching-based nature of STCN and includes mask propagation via a memory of masks, while the *tt*-AE variant does not include the matching step between multiple frames.

Given a test video, we optimize the parameters of the model using the provided mask and either only the first frame (for *tt*-AE) or all its frames (*tt*-Ent, *tt*-MCC). The test-time loss is optimized, the model parameters are updated, and the model is used to provide predictions for the specific test video. Then, the model is re-initialized back to its original parameters before another test video is processed.

For the two TTT variants that use multiple frames, *i.e.* *tt*-MCC and *tt*-Ent, we follow the process used during the off-line training phase of STCN and XMem and operate on randomly sampled frame triplets and octuplets, respectively. In the following, we describe the case of triplets which is extended to larger sequences in a straightforward way. All the triplets include the first video frame, while the second (third) triplet frame is uniformly sampled among the $s$ frames that follow the first (second) triplet frame in the video. For triplet $x_0, x_i, x_j$ with $0 < i < j$, the first predicted mask is given by $\hat{m}_i = f(x_i; \{(x_0, m_0)\})$, and the second by $\hat{m}_j = f(x_j; \{(x_0, m_0), (x_i, \hat{m}_i)\})$, *i.e.* the second frame is added to the memory before predicting the last mask of the triplet. Multiple triplets are sampled during test-time training. The per-pixel losses are averaged to form the frame/mask loss. We denote the cross entropy loss between a ground-truth mask $m$ and a predicted mask $\hat{m}$ by $\ell_{CE}(m, \hat{m})$.

### 3.2.1 Entropy (*tt*-Ent) loss

We start from a task- and method-agnostic loss based on Entropy (*tt*-Ent) minimization, a common choice for test-time training on image classification [46]. In this case, we force the model to provide confident predictions for the triplets we randomly sample. The entropy of mask $\hat{m}_i$ is denoted by $H(\hat{m}_i)$ and the loss for a particular triplet is $L_H = 0.5 * (H(\hat{m}_i) + H(\hat{m}_j))$, which is minimized over multiple triplets. Prior work on image classification avoids the trivial solution by using a batch of test examples [46] or augmenting the input example and minimizing the entropy over the average prediction of the augmented frames [59]. In the task of semantic segmentation, batches or augmentations are not required since it is a dense prediction task and we have multiple outputs to optimize, *i.e.* a prediction per pixel. In the case of VOS, we have even more outputs due to the temporal dimension and because we are sampling triplets among many frames. However, similar to Wang et al. [46], we observe that training only the batch normalization parameters is a requirement for *tt*-Ent to work.

### 3.2.2 Mask auto-encoder (*tt*-AE) loss

The STCN model includes a mask encoder and a mask decoder to decode the mask representation coming from the cross-attention process. We repurpose these components for test-time training and compose an auto-encoder that reconstructs the input mask. The auto-encoder input is the provided ground-truth mask, *i.e.* $m_0$, and the loss is given by $L_{AE} = \ell_{CE}(m_0, d_m([e_m(m_0), e_x(x_0)]))$. Note that, together with the mask encoder that gets optimized to better represent the specific object shape, we also optimize the frame encoder with the *tt*-AE loss, since the mask representation is concatenated with the frame representation in the input of the mask encoder [9].

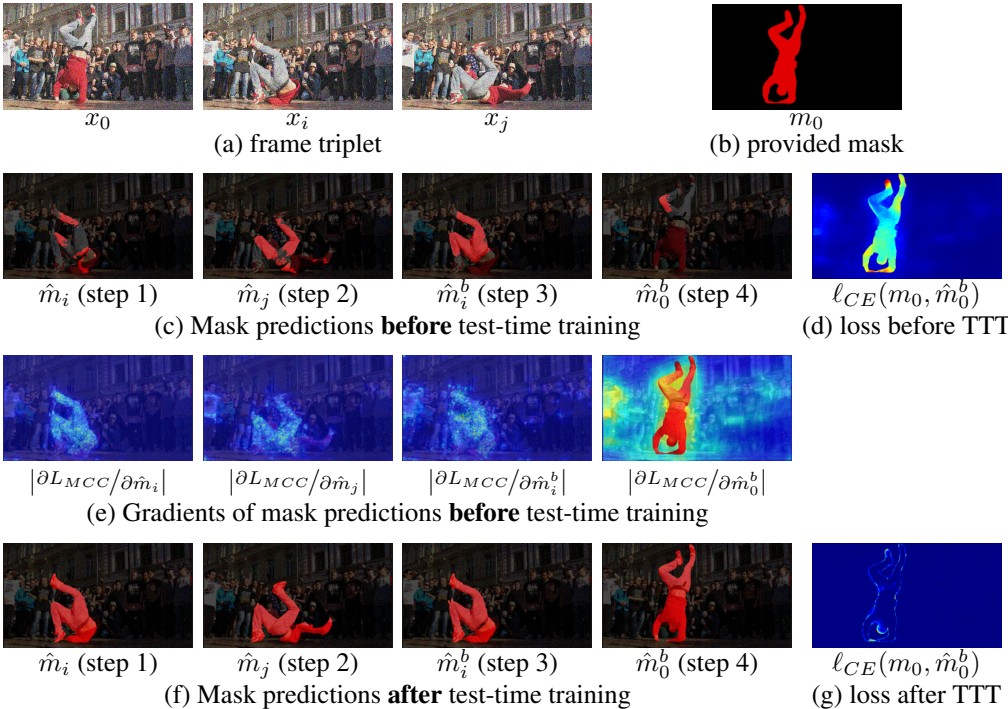

Figure 3: **Unravelling the** $tt$**-MCC loss.** (a) a triplet from a video with gaussian noise corruption; (b) the provided mask; (c) mask predictions after each of the four steps of the cycle, before test-time training (TTT); (d) the pixel-wise loss before TTT; (e) gradients of mask predictions for each of the 4 steps of the cycle; (f) predictions after test-time training with the mask cycle consistency loss; (f) the loss after TTT. We see that $tt$-MCC significantly improves all predicted masks and reduces the loss. We binarize the predicted masks for visualization purposes; they are otherwise probability maps.

### 3.2.3 Mask cycle consistency ($tt$-MCC) loss

After sampling a triplet, we make predictions in a time-forward and time-backward manner to form a cycle that allows us to use the provided ground-truth mask as a label with a cross-entropy loss. The cycle consists of four steps. We first obtain masks $\hat{m}_i$ and $\hat{m}_j$ with time-forward predictions. Then, we consider the triplet in reverse order going from $x_j$ to $x_i$ and then to $x_0$, and obtain time-backward predictions. In particular, $\hat{m}_j$ is treated as the only mask in the memory to enable prediction for frame $x_i$ which is given by $\hat{m}_i^b = f(x_i, \{(x_j, \hat{m}_j)\})$. In the last step, the mask of $x_0$ is given by $\hat{m}_0^b = f(x_0, \{(x_j, \hat{m}_j), (x_i, \hat{m}_i^b)\})$. We compare this prediction to the provided ground-truth mask and the loss to optimize is $L_{MCC} = \ell_{CE}(\hat{m}_0^b, m_0)$. The overall process and the four prediction steps are shown in Figure 2. During the optimization, back-propagation is performed through all the predicted masks mentioned above. In this way, earlier predictions are also optimized in an indirect way.

For successful object segmentation in $\hat{m}_0^b$, masks $\hat{m}_i$, $\hat{m}_j$, and $\hat{m}_i^b$, in the whole sequence of predictions, are required to have successful object segmentation too. If for example an object part is not properly segmented in the intermediate predictions, it will be missed in $\hat{m}_0^b$ too. We further experimented with adding an extra mask consistency loss between the forward and backward predictions for the middle/intermediate frame, *i.e.* $\ell_{CE}(\hat{m}_i^b, \hat{m}_i)$ but did not result in any additional gains.

Figure 3 shows the four predictions during a cycle using a particular triplet before and after $tt$-MCC. We see that initially, the object is not properly segmented during internal steps of the cycle and is fully missed in the last step, of the cycle resulting in large loss values for most of its pixels. After TTT is performed, not only the mask which is given as input to the loss, *i.e.* $\hat{m}_0^b$ gets improved, but so are the predictions during intermediate steps. Note that the prediction of step 4 is not part of the inference process, but is rather a proxy to optimize predictions for steps 1 and 2 that are part of it. The gradients of the internal masks are indicating how the loss is affecting their predictions per pixel.

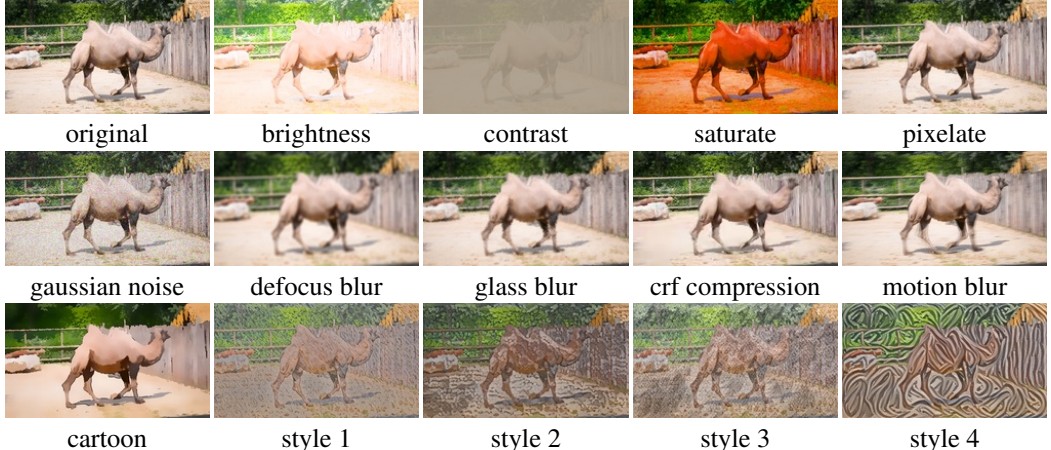

Figure 4: **The 14 types of corruption in DAVIS-C**. Corruptions are depicted at "medium" strength.

# 4 Experiments

In this section, we first describe the training and implementation details. Then, we introduce the DAVIS-C benchmark and report our experimental results on four datasets and two distribution shifts: sim-to-real transfer and corrupted/stylized test data.

**Initial models.** We start from two publicly available models for STCN. STCN-BL30K is the model trained on synthetic data from BL-30K [8]. STCN-DY is a model initialized by STCN-BL30K and further trained using the training videos of DAVIS and YouTube-VOS, which include approximately 100K frames annotated with segmentation masks. We use the same two models for XMem, denoted by XMem -BL30K and XMem -DY, respectively.

**Training details.** We use learning rates $10^{-5}$ and $10^{-6}$ for models STCN-BL30K/ XMem -BL30K and STCN-DY/ XMem -DY, respectively since their training data differ significantly. Jump step $s$ for sampling training frames is set to 10. For each test example, we train the models with $tt$-MCC and $tt$-Ent for 100 iterations and with $tt$-AE for 20, using the Adam [23] optimizer and a batch size of 4 sequences for STCN and 1 for XMem. We consider all target objects in the ground truth mask during TTT, and the raw videos are used with no augmentations. We develop our methods on top of the publicly available STCN[2] and XMem[3] implementations.

## 4.1 Datasets and evaluation

We report results on the two most commonly used benchmarks for video object segmentation, the DAVIS-2017 validation set [38] and the YouTubeVOS-2018 validation set [51]. We further report results on the validation set of the recent MOSE [10] dataset, which includes challenging examples with heavy occlusion and crowded real-world scenes. Finally, we introduce DAVIS-C, a variant of the DAVIS test set that represents distribution shifts via corruptions and stylization. We evaluate all methods using the widely established $\mathcal{J}\&\mathcal{F}$ measure similar to prior work [9, 6].

**DAVIS-C: corrupted and stylized DAVIS.** To model distribution shifts during the testing, we process the set of test videos of the original DAVIS dataset to create DAVIS-C. This newly created test set offers a test bed for measuring robustness to corrupted input and generalization to previously unseen appearance. It is the outcome of processing the original videos to introduce image-level and video-level corruption and image-level appearance modification. We perform 14 different transformations to each video, each one applied at three different strengths, namely *low*, *medium*, and *high* strength, making it a total of 42 different variants of DAVIS.

---

[2]https://github.com/hkchengrex/STCN
[3]https://github.com/hkchengrex/XMem

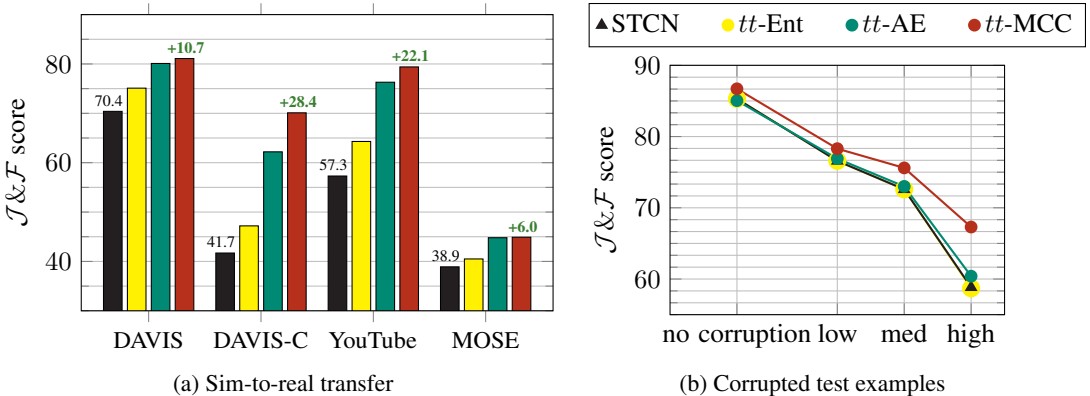

| (a) Sim-to-real transfer | (b) Corrupted test examples |

Figure 5: **VOS performance under distribution shifts.** *Left:* performance of STCN-BL30K before and after test-time training for the sim-to-real transfer case on four datasets. *Right:* performance of STCN-DY on DAVIS-C for input corruptions with different strength levels.

For the image-level corruptions, we follow the paradigm of ImageNet-C and adopt 7 of their corruptions, namely gaussian noise, brightness, contrast, saturate, pixelate, defocus blur, and glass blur. The different transformation strengths are obtained by using severity values 3, 4, and 5 from ImageNet-C. For the video-level corruptions, we introduce constant rate factor (CRF) compression for three increasing values of compression and motion blur by averaging every 2, 3, and 4 consecutive frames. For the image-level transformations, we create cartoonization using the mean shift algorithm by increasing the amount of smoothing and image-stylization [13] using four different styles. For stylization, we do not vary the strength, but we preserve the original video colors (low and medium strength) while switching to those of the style image (high strength), making it a more drastic appearance shift. Frame examples for all cases are shown in Figure 4.

Some of the transformations in DAVIS-C do not perfectly represent video distortions. Nevertheless, several of the transformations constitute common real-world video edits (contrast, brightness, crf compression, cartoonization, stylization). We believe DAVIS-C provides a valuable benchmark to study video understanding under distribution shift.

### 4.2 Results

**Sim-to-real transfer.** In Figure 5a, we report results with and without test-time training on four datasets for the case of sim-to-real transfer. All three TTT variants bring gains consistently across datasets, with $tt$-MCC improving performance significantly, *i.e.* a relative gain of more than 39%, 15% and 15% for YouTube-VOS, DAVIS and MOSE, respectively. Moreover, we see that naively applying task-agnostic entropy minimization test-time training brings only a small percentage of the gains one can get. It is also worth noting that the gains from TTT are so high that performance using the mask cycle consistency variant becomes comparable to the performance of a model trained on video data. Specifically, $tt$-MCC recovers 82%, 72% and 44% of the performance gains that training with all the training videos from YouTube-VOS and DAVIS combined brings. This is very promising in cases where annotated data are scarce but synthetic data is widely available.

**Corrupted test examples.** We report results for DAVIS-C in Figures 5b, 6, and the right side of Table 1. Figure 5b clearly shows how $tt$-MCC outperforms STCN for all corruption levels, with the gains increasing more as corruption strength increases. The entropy minimization approach does not offer any gains, while $tt$-AE is effective but worse than $tt$-MCC. In Figure 6, we report the $\mathcal{J}\&\mathcal{F}$ score separately per video of DAVIS-C at the highest strength. We see that the gains come from many transformations and examples, while we see that TTT is able to improve most videos where the original $\mathcal{J}\&\mathcal{F}$ score was in the lower side of the spectrum. Finally, on the right side of Table 1, we further report results on DAVIS-C for XMem and see that gains persist for one more state-of-the-art method. Interestingly, from Table 1, we also see that the performance gain of XMem over STCN disappears for the sim-to-real transfer case for YouTube-VOS, as well as for the extreme case a model trained only on synthetic videos is tested on corrupted inputs.

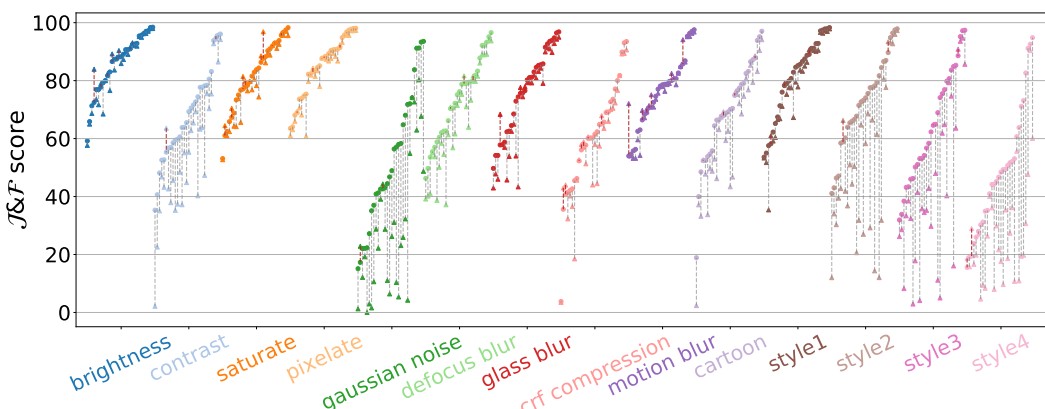

Figure 6: **Test-time training on DAVIS-C.** We plot the $\mathcal{J}\&\mathcal{F}$ score separately per video before (triangles) and after $tt$-MCC (circles) for the STCN-DY model on the 14 corruptions of the proposed DAVIS-C benchmark. We report results for the variants with the highest corruption strength. A red (grey) vertical line denotes that performance drops (increases) with test-time training.

Table 1: **Comparison to the state-of-the-art** for two cases of test-time distribution shift. Left part: Results when using models trained without real videos. The HODOR model is trained on COCO [27] images, while STCN and XMem on BL-30K. Right part: Results on DAVIS-C for different levels of corruption. HODOR, STCN and XMem are fine-tuned with DAVIS and YouTube-VOS videos. [†] Results from [1].

| Method | Training without real videos | | | | Corrupted test examples (DAVIS-C) | | | | |
|---|---|---|---|---|---|---|---|---|---|
| | DAVIS | DAVIS-C | YouTube-VOS | MOSE | no corr. | low | med | high | avg |
| HODOR [1] | 77.5[†] | – | 71.7[†] | – | 81.3[†] | 69.0 | 64.5 | 55.2 | 65.0 |
| STCN [9] | 70.4 | 41.7 | 57.3 | 38.9 | 85.3 | 76.6 | 72.6 | 58.8 | 73.3 |
| STCN + $tt$-MCC (ours) | 81.1 | **70.1** | **79.4** | **44.9** | 86.7 | 78.3 | 75.6 | 67.3 | 77.0 |
| XMem [6] | 78.1 | 53.9 | 65.6 | 40.9 | 87.7 | 80.4 | 77.3 | 69.4 | 78.7 |
| XMem + $tt$-MCC (ours) | **82.1** | **70.1** | 78.9 | 44.7 | **88.1** | **81.7** | **78.9** | **72.2** | **80.2** |

**Comparison to the state of the art.** In Table 1, we additionally report results for our $tt$-MCC variant when starting from the state-of-the-art XMem [6] method. Once again, the gains are consistent, for both cases. It is worth noting that from the fourth column of the table, we can see that $tt$-MCC is also able to slightly improve the case when there is no test-time distribution shift, *i.e.* we report 1.4 and 0.4 higher $\mathcal{J}\&\mathcal{F}$ score on DAVIS for STCN-DY and XMem -DY, respectively. In the table, we also compare our method to HODOR [1] method that uses cycle consistency during the offline training stage and reports results when training without real videos. Our STCN-BL30K model with TTT outperforms HODOR by **+3.6** and **+8.3** $\mathcal{J}\&\mathcal{F}$ score, on DAVIS and YouTube-VOS respectively.

**Impact of sampling longer sequences.** Regarding the sequence length used during test-time training, we adopt the setup used by the base methods during their training phase, *i.e.* frame triplets for STCN and octuplets for XMem. We experiment with quadruplets or quintuplets for mask cycle consistency on top of STCN which achieve performance of 81.7 and 82.4, respectively, on DAVIS versus 81.1 for triplets. This gain in performance, however, comes with a significant increase in test-time training time, *i.e.* 33% and 66%, respectively.

**Impact of changing the sampling strategy.** Additionally to changing the sequence length, another way to affect the size of the temporal context is by the value of the jump step, *i.e.* the interval used to sample frames $x_i$ and $x_j$ for a training triplet. Test-time training with $tt$-MCC over the STCN-BL30K model achieves 81.1 on DAVIS, with the jump step $s$ parameter set to 10 by default. Varying the jump step from 1, 2, 25, and 50, achieves 80.6, 81.0, 80.8, and 79.0, respectively. Neither sampling too close to the first frame nor sampling frames too far from each other is the optimal choice. Switching to random sampling, without any step as a restriction, causes a performance drop to 79.7, while sampling only within the last 5% or 10% of the video also causes a drop to 79.8 and 79.7, respectively.

Across all strategies and hyperparameter values, we see noticeable improvements using $tt$-MCC; the base model without TTT, merely achieves 70.4 $\mathcal{J}\&\mathcal{F}$ score on DAVIS.

**Impact of training iterations.** For all results we use a fixed number of 100 iterations during TTT. In Figure 7, we present the performance improvement of $tt$-MCC for all 30 DAVIS videos (one curve per video) for a varying number of iterations. One can clearly see that optimal performance is achieved at different iterations per video, and we believe that adaptive early stopping is a promising future direction that may boost the overall improvement even further.

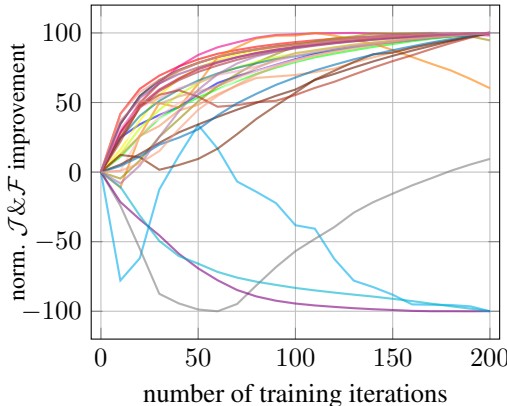

Figure 7: **Change in $\mathcal{J}\&\mathcal{F}$ score during TTT** compared to the score before TTT. Each curve is one video and is normalized wrt. the maximum improvement/decline observed over 200 iterations.

**Impact on inference time.** We discuss the effect that test-time training has on inference speed, by measuring average inference time over all DAVIS validation videos. The vanilla STCN model requires roughly 2 seconds per video. For $tt$-MCC, $tt$-Ent, and $tt$-AE, it takes 67, 29, and 5.3 seconds, respectively, for 100 iterations. Since TTT is run per video and its speed is only affected by the number of iterations, per frame timings vary with video length, *i.e.* the TTT overhead reduces as video length increases. It is worth noting here that all timings presented above are estimated with a non-optimized TTT implementation that leaves a lot of space for further optimization.

**Reducing TTT overhead.** As discussed, TTT induces a significant overhead at test time, which is an aspect not often discussed in the relevant literature. The TTT overhead is reduced by training for less iterations with only a minor decrease in performance. As we show Table 3 of the Appendix, using 20 instead of 100 iterations is even advantageous in cases, particularly for larger datasets like YouTube-VOS and MOSE with a model trained on real videos and without a distribution shift. In a real-world application, applying $tt$-MCC on all test examples might seem infeasible due to time overheads. Nevertheless, it is very useful for cases of extreme test-time distribution shifts. In scenarios involving a few out-of-distribution examples, where rapid adaptation of the current model is desired for improved performance, $tt$-MCC constitutes a straightforward and cost-effective solution, that is far more efficient than retraining the base model, which typically requires approximately 12.5 hours on 2 A100 GPUs to train STCN.

**Memory requirements.** During inference, STCN requires roughly 8GB of GPU RAM. Test-time training on top of STCN with $tt$-AE, $tt$-Ent and $tt$-MCC requires 8,16 and 23.5GB, respectively, *i.e.* it can still fit in a modest GPU. Note that the calculations for TTT are computed when using a batch size of 4 (the default). Memory requirements can be further reduced by using a smaller batch size, if needed, without significant change in performance.

## 5 Conclusions

In this work we show that test-time training is a very effective way of boosting the performance of state-of-the-art matching-based video object segmentation methods in the case of distribution shifts. We propose a mask cycle consistency loss that is tailored to matching-based VOS and achieves top performance. Its applicability goes beyond the two methods used in this work, as it is compatible with the general family of matching-based VOS methods [36, 60, 56, 1]. We report very strong gains over top performing methods for both sim-to-real transfer and the case of corrupted test examples. We also show that achieving such gains is not trivial and that it is important to tailor the test-time training method to the task and method at hand. A limitation of the proposed approach is the lack of a way for performing early stopping, and selecting the best iteration to stop training, a very promising direction for future work.

## Acknowledgments and Disclosure of Funding

This work was supported by Naver Labs Europe, by Junior Star GACR GM 21-28830M, and by student grant SGS23/173/OHK3/3T/13. The authors would like to sincerely thank Gabriela Csurka for providing valuable feedback.

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

# Appendix

## Contents

## A   Datasets and additional implementation details

### A.1   Datasets

**DAVIS**   The validation split of the DAVIS-2017 [38] dataset contains 30 videos covering a variety of real-world scenarios, including indoor and outdoor scenes, different lighting conditions, occlusions, and complex motion patterns. Each video contains one to three annotated objects of interest.

**YouTubeVOS-2018**   The validation split of the YouTubeVOS-2018 [51] dataset contains 474 high-quality videos downloaded from YouTube, including indoor and outdoor scenes, different lighting conditions, occlusions, and complex motion patterns. Each video contains one to five annotated objects of interest.

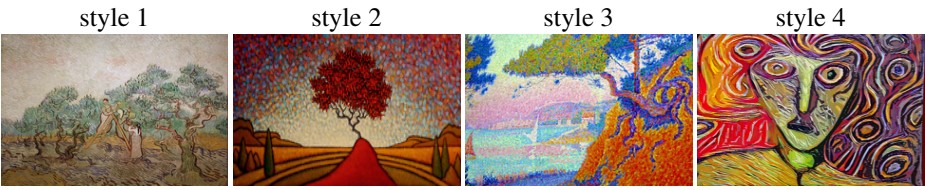

Figure 8: **The four styles in DAVIS-C**.

Table 2: **Multiple runs of the proposed method on top of STCN and XMem** for two cases of test-time distribution shift. Mean and standard deviation over 3 different seeds is reported. Left part: Results for STCN-BL30K and XMem-BL30K. Right part: Results for STCN-DY and XMem-DY on DAVIS-C for different levels of corruption.

| Method | Training without real videos | | | | Corrupted test examples (DAVIS-C) | | | | |
|---|---|---|---|---|---|---|---|---|---|
| | DAVIS | DAVIS-C | YT-VOS | MOSE | no corr. | low | med | high | avg |
| STCN [9] | 70.4 | 41.7 | 57.3 | 38.9 | 85.3 | 76.6 | 72.6 | 58.8 | 73.3 |
| STCN + $tt$-MCC (ours) | $81.1_{\pm0.1}$ | $70.1_{\pm0.1}$ | $\mathbf{79.4}_{\pm0.2}$ | $\mathbf{44.9}_{\pm0.2}$ | $86.7_{\pm0.2}$ | $78.3_{\pm0.1}$ | $75.6_{\pm0.1}$ | $67.3_{\pm0.1}$ | 77.0 |
| XMem [6] | 78.1 | 53.9 | 65.6 | 40.9 | 87.7 | 80.4 | 77.3 | 69.4 | 78.7 |
| XMem + $tt$-MCC (ours) | $\mathbf{82.1}_{\pm0.2}$ | $\mathbf{70.1}_{\pm0.3}$ | $78.9_{\pm0.2}$ | $44.7_{\pm0.2}$ | $\mathbf{88.1}_{\pm0.2}$ | $\mathbf{81.7}_{\pm0.1}$ | $\mathbf{78.9}_{\pm0.2}$ | $\mathbf{72.2}_{\pm0.1}$ | **80.2** |

**DAVIS-C** We refer the reader to the main paper for details. Here, we highlight the 14 different transformations to each video applied at three different strengths, namely *low*, *medium*, and *high* strength in Figure 17. The four original images that we use to compute the four stylizations *style 1*, *style 2*, *style 3*, and *style 4* are displayed in Figure 8.

**MOSE** We further report results on a newer, much larger dataset, *i.e.* the MOSE dataset [10], which contains challenging examples with heavy occlusions and crowded real-world scenes. The validation split of the MOSE dataset contains 311 high-quality videos. Each video contains one to fifteen annotated objects of interest. In addition, to show the robustness of our method, we also ran our models on the training split of the MOSE dataset, which contains 1507 high-quality videos with up to twenty annotated objects of interest. Because the STCN and XMem models are not trained on the MOSE-train split before, we consider this split as a larger dataset for testing our method. Because of limited space, we present results on MOSE-train only in the supplementary, and as we show in Section B, the observations of the main paper also hold in this case.

### A.2 Additional implementation details

The model is in evaluation mode during test-time training, *i.e.* the running statistics of the Batch Normalization layers remain fixed.

Because the last frame $m_j$ of the frame sequence used in our $tt$-MCC method is the initial frame of the backward pass, all the target objects in $m_0$ must also be depicted in $m_j$. To guarantee this, we select only the frames where the network can detect all the target objects, *i.e.* there is at least one pixel for each object where it has the largest probability. We do so by extracting pseudo masks for the video frames and resetting the optimizer every ten iterations.

In the MOSE dataset, some videos have up to twenty annotated objects. Because our method is linear with the number of annotated objects in the video, we randomly select up to six objects at each iteration.

## B Additional experimental results and analysis

### B.1 Measuring the standard deviation of TTT

We report mean and standard deviation after running TTT with 3 different seeds in Table 2 for four datasets, namely DAVIS, DAVIS-C, YouTube-VOS, and MOSE.

### B.2 Extended results on the MOSE dataset

In Figure 9, we report STCN results with and without test-time training on the two splits of the MOSE dataset for the case of sim-to-real transfer. We can see that on both splits of the MOSE dataset, the three TTT variants boost the performance, and $tt$-MCC brings a relative gain of 15% on 18% on the MOSE-valid and MOSE-train datasets, respectively.

### B.3 Results with TTT on top of XMem

In Figure 10, we show the performance comparison with and without TTT on top of XMem. We observe that the performance benefits are similar to those of using TTT on top of STCN.

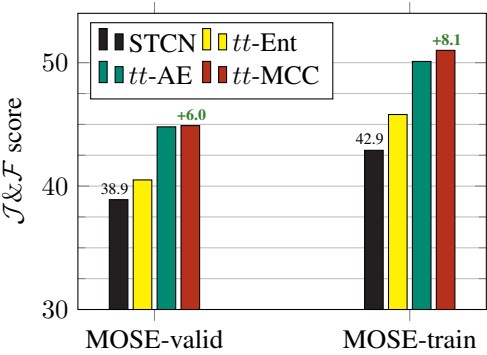

Figure 9: **STCN performance on the MOSE dataset for the sim-to-real transfer case.** We report results on both the validation set, as well as the much larger training set of 1507 videos; used as another test set since it is not used as a training set by the models used in this work.

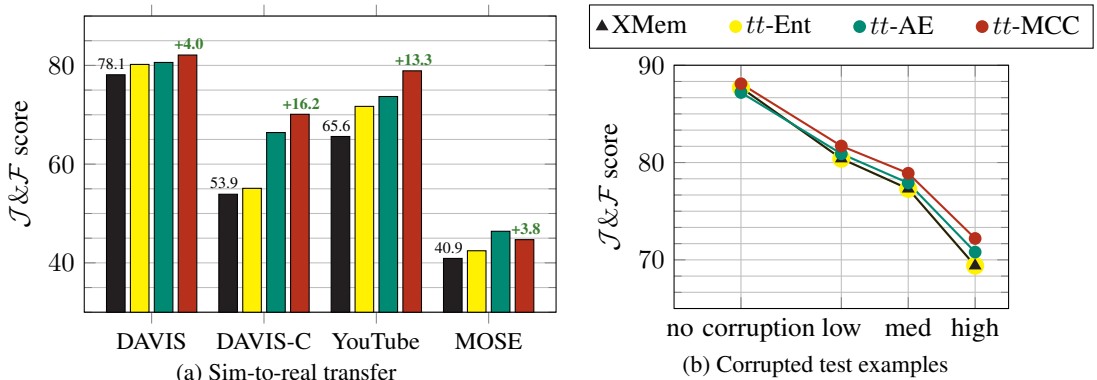

(a) Sim-to-real transfer

(b) Corrupted test examples

Figure 10: **XMem performance under distribution shifts.** *Left:* performance of XMem -BL30K before and after test-time training for the sim-to-real transfer case on four datasets. *Right:* performance of XMem -DY on DAVIS-C for input corruptions with different strength levels.

## B.4 Varying the number of real video used for training the base model

We increase the number of real videos used to train the STCN model from 0 (BL-30K) to 3531 (DAVIS and YouTube-VOS training sets) and present results in Figure 11. When the model is trained with little to no real videos, test-time training recovers the bulk of the performance achieved using models trained on larger annotated datasets while requiring little to no annotation.

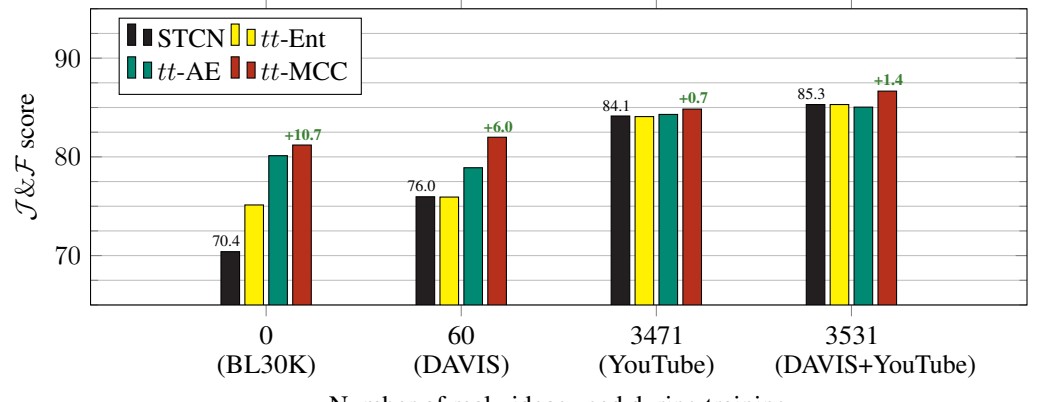

Number of real videos used during training

Figure 11: **Performance on DAVIS with respect to the number of real videos used during training.** Performance before and after test-time training for models trained using only synthetic videos (BL-30K) and up to 3531 real videos (DAVIS and YouTube-VOS together).

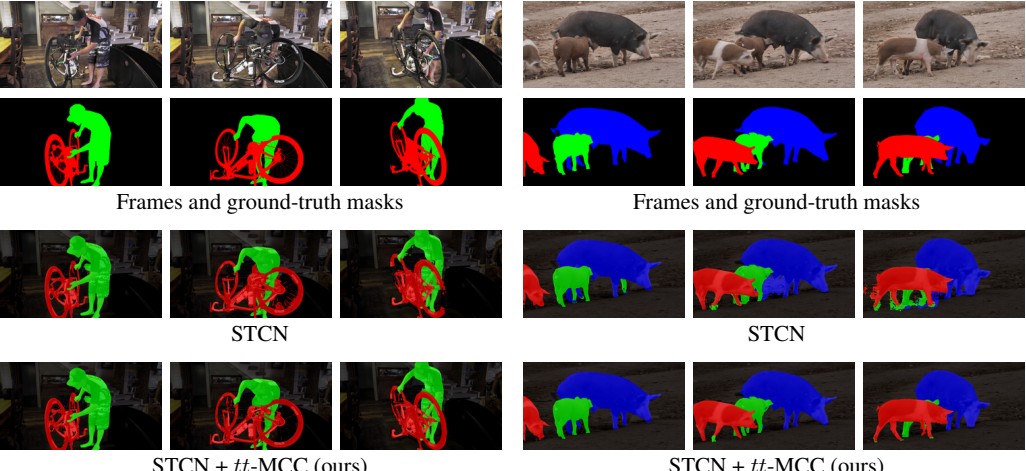

Figure 12: **Additional qualitative examples using a model trained on synthetic videos** from BL-30K [8] and tested on a real video from DAVIS [38]. Second-to-bottom row: Results obtained using the STCN [9] approach. Bottom row: Results after test-time training using the proposed *mask cycle consistency* loss ($tt$-MCC) on the single ground-truth mask provided for the first video frame.

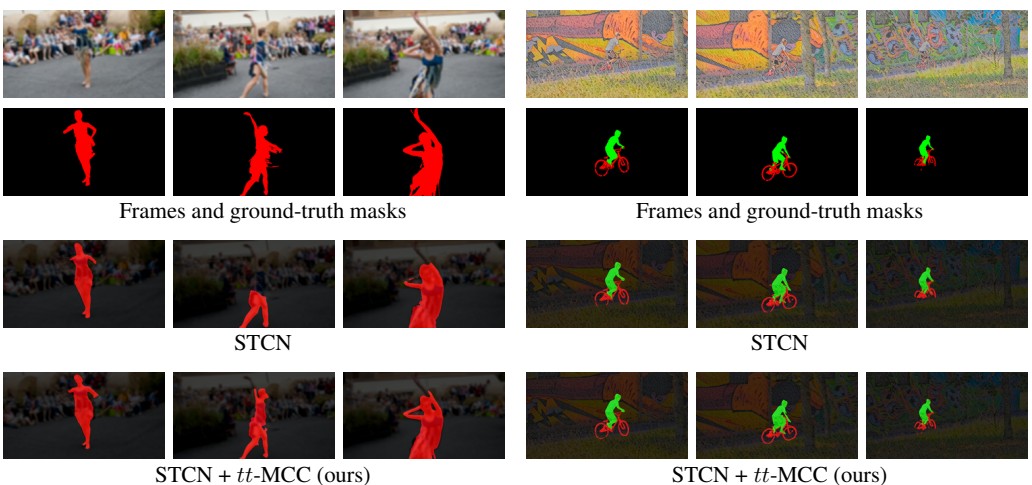

Figure 13: **Additional qualitative examples using a model trained on DAVIS [38] and YouTube-VOS [51] and tested on a corrupted video from the DAVIS-C benchmark.** *Left:* the test example is corrupted using the *glass-blur* corruption with medium strength. *Right:* the test video is stylised using *style 1* with medium strength. Second-to-bottom row: Results obtained using the STCN [9] approach. Bottom row: Results after test-time training using the proposed *mask cycle consistency* loss ($tt$-MCC) on the single ground-truth mask provided for the first video frame.

## B.5   Additional qualitative results

We show more qualitative examples in Figures 12 and 13.

## B.6   Results in the case of no distribution shift

In the main part of the manuscript, we focus on scenarios involving test-time distribution shifts. There, we report results based on 100 iterations, which we've found to be optimal for all cases with such distribution shifts. When a model is trained on real videos and there is no distribution shift in the test videos, as demonstrated in the left part of Table 3, using fewer iterations is advantageous, particularly for larger datasets like YouTube-VOS and MOSE. Overall, we observe that $tt$-MCC does

Table 3: **Additional results for STCN and Xmem**. The number of TTT iterations is reported next to *tt*-MCC. Left part: Results when starting from a model trained with real videos. Right part: Results when starting from a model trained only on static images.

| Method | Training with real videos | | | | Training with static images | | | |
|---|---|---|---|---|---|---|---|---|
| | DAVIS | DAVIS-C | YT-VOS | MOSE | DAVIS | DAVIS-C | YT-VOS | MOSE |
| STCN | 85.3 | 69.3 | 84.3 | 52.5 | 75.7 | 59.3 | 76.3 | 42.0 |
| STCN + *tt*-MCC-20 (ours) | 86.0 | 72.1 | 84.6 | 53.3 | 78.7 | 68.8 | 78.2 | **43.6** |
| STCN + *tt*-MCC-100 (ours) | 86.7 | 73.8 | 84.0 | 51.8 | **79.7** | **70.7** | **79.4** | 43.4 |
| XMem | 87.7 | 75.7 | **86.1** | 60.4 | 72.8 | 56.2 | 77.0 | 42.3 |
| XMem + *tt*-MCC-20 (ours) | 88.0 | 76.8 | 85.8 | **60.7** | 75.9 | 63.0 | 76.5 | 41.7 |
| XMem + *tt*-MCC-100 (ours) | **88.1** | **77.6** | 85.1 | 59.8 | 78.8 | 67.2 | 78.4 | 42.9 |

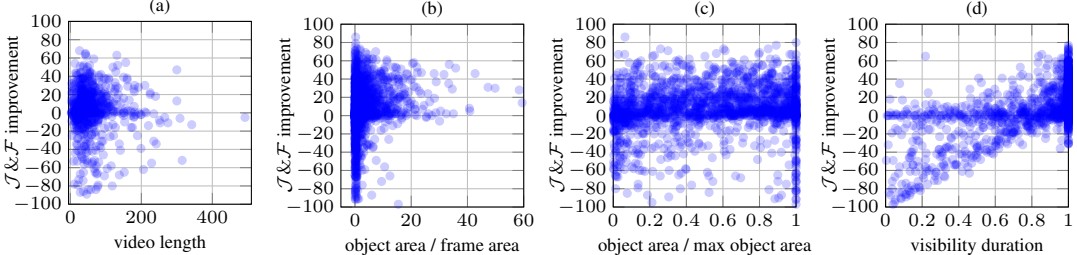

Figure 14: **Performance gain of tt-MCC for the sim-to-real case on the MOSE-train dataset.** We plot the performance gain *vs*. a) the video length in number of frames, b) the object area in the first frame normalized by the frame area, c) the object area in the first frame normalized by the maximum object area over all frames, d) the percentage of the video length where the object is visible.

not negatively impact the performance of state-of-the-art methods in these cases. Instead, it offers a method to either maintain or enhance state-of-the-art performance across all test-time video scenarios, whether extreme distribution shifts are present or not.

### B.7 Results in the case of training with static images

In this section, we focus on the scenario where the model is trained using only static images. Following STCN, we employ five datasets of static images [47, 41, 58, 7, 25] for the offline training of the networks. Given a training image, several deformed versions are generated to compose an artificial video sequence, which is used for the model training. The corresponding results are presented in the right part of Table 3 for both STCN and XMem across the four datasets. Notably, for STCN, *tt*-MCC yields relative performance gains exceeding 5%, 4%, and 3% on DAVIS, YouTube-VOS, and MOSE, respectively. These gains represent substantial recovery of the performance achieved by a model trained with real videos, *i.e.* amounting to 42%, 39%, and 13%, respectively. Performance improvement is even more significant on the DAVIS-C dataset, with a 19% relative performance gain, surpassing the STCN model trained on real videos (70.7 *vs*. 69.3).

### B.8 Performance analysis on a per-video and per-object basis

We conduct a comprehensive analysis using the MOSE-train dataset on a per-video and per-object basis and present results in Figure 14. Figure 14 (a) examines the performance gain using TTT in relation to video length, revealing that video length does not significantly affect performance. Figure 14 (b) demonstrates a subtle correlation between performance gain and the size of objects in the first frame, with larger objects showing positive gains and no negative impact. In Figure 14 (c) , we explore how the object's size in the initial frame relative to its maximum size in the video affects performance. We observe a slight negative impact when the object is less visible in the first frame compared to subsequent frames. This is attributed to *tt*-MCC overfitting to the partially annotated first frame and often segmenting the partial object in future frames. Note that extreme cases of severe occlusion in the first frame can make the video object segmentation task particularly ambiguous. Finally, Figure 14 (d) assesses the impact of object visibility duration within the video sequence,

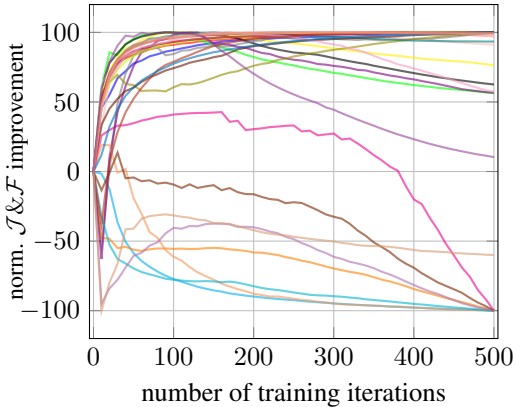

Figure 15: **Change in $\mathcal{J}\&\mathcal{F}$ score during TTT** compared to the score before TTT. Each curve is one video and is normalized wrt. the maximum improvement/decline observed over 500 iterations.

confirming that $tt$-MCC is adversely affected when the object is briefly visible. This suggests that our cycle consistency loss, emphasizing long-term temporal consistency, may introduce bias towards objects being visible for extended periods.

### B.9 Impact of longer test-time training

To better study the effect of longer training at test-time, we run our method for up to 500 iterations on DAVIS for the sim-to-real case and presents results in Figure 15. In summary, we do not observe any significant further decrease in performance nor any degenerate outputs after longer training. More specifically, the number of DAVIS videos for which VOS performance decreases remained the same from 50 TTT iterations up to approximately 400, *i.e.* the $\mathcal{J}\&\mathcal{F}$ score decreases in 7 out of 30 videos. Only one extra video shows a decrease in $\mathcal{J}\&\mathcal{F}$ score by 500 iterations. Visually observing segmentation results, we notice that a common issue is some background pixels of similar appearance to the object are wrongly segmented. It is also worth noting that increasing the iterations only leads to a minor overall decrease in VOS performance, *i.e.* $\mathcal{J}\&\mathcal{F}$ score drops from 81.1 to 81.0/80.4 for 200/500 iterations, respectively.

### B.10 Assessing temporal stability

In Figure 16, we present a performance comparison on a per-video basis, both with and without $tt$-MCC. The graph illustrates consistent and lasting performance improvements. Importantly, it highlights the enhanced stability of performance when $tt$-MCC is employed. In the case of a briefly visible object (as seen in video `3bfa8379`), our use of $tt$-MCC results in incorrectly segmenting out the background as the object, significantly impacting the metric in a negative manner.

### B.11 Detailed results on DAVIS-C

In Figures 18 and 19, we present the $\mathcal{J}\&\mathcal{F}$ score change before and after test-time training for the STCN-BL30K and STCN-DY models, respectively, for all three corruption strengths.

### B.12 Failure cases

We notice a couple of patterns when inspecting failure cases of our method, and TTT in general. The most important is that TTT might make the model more confident about a dubious prediction and propagate it wrongly. Moreover, we observe that TTT might overfit the appearance module and wrongly add background pixels. *E.g.* we observe that a red shirt from the background in the breakdancer video confuses the model more than the STCN case. We refer the reader to the video we provide on the project webpage, where we visualize a number of such failure cases.

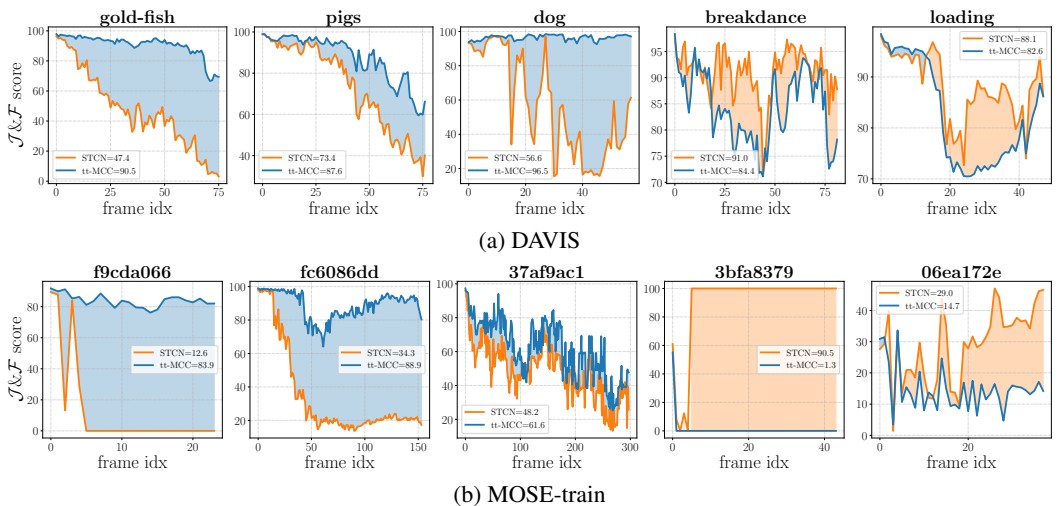

(a) DAVIS

(b) MOSE-train

Figure 16: **Performance per frame of STCN and our $tt$-MCC for the sim-to-real case on selected videos from DAVIS and MOSE-train.** We plot the performance of each method on each frame of the test video averaged over all objects. The colors of the highlighted areas indicate the winning method. The average performance for the whole video is reported in the legend per method.

## C   Broader impact

The proposed method is effective for the two types of distribution shifts evaluated in our work. We expect it to be valuable for knowledge transfer to very different domains with limited or no annotated data. For instance, in the bio-medical domain, for a task such as tracking cells or organisms captured under a microscope. Due to the lack of appropriate datasets, we do not evaluate the proposed approach in such a use case.

Test-time training methods like ours can also be used to improve the semi-automatic annotation process for large video collections in the presence of distribution shifts. Such data can then be used to train and fine-tune other models in other domains and tasks. It is worth noting that whatever the annotation, societal, or otherwise biases existing in the dataset the base model was trained on will also be present in the automatically annotated frames and propagated through this process. Test-time training on a single annotated example is not expected to successfully correct such biases; understanding the limitations of the base model is therefore crucial for any real-world application of our method.

## D   Figures for all TTT losses

We illustrate the $tt$-Ent and $tt$-AE losses, as well as the regular cross-entropy loss for the supervised offline training stage in Figures 20a, 20b, and 20c, respectively.

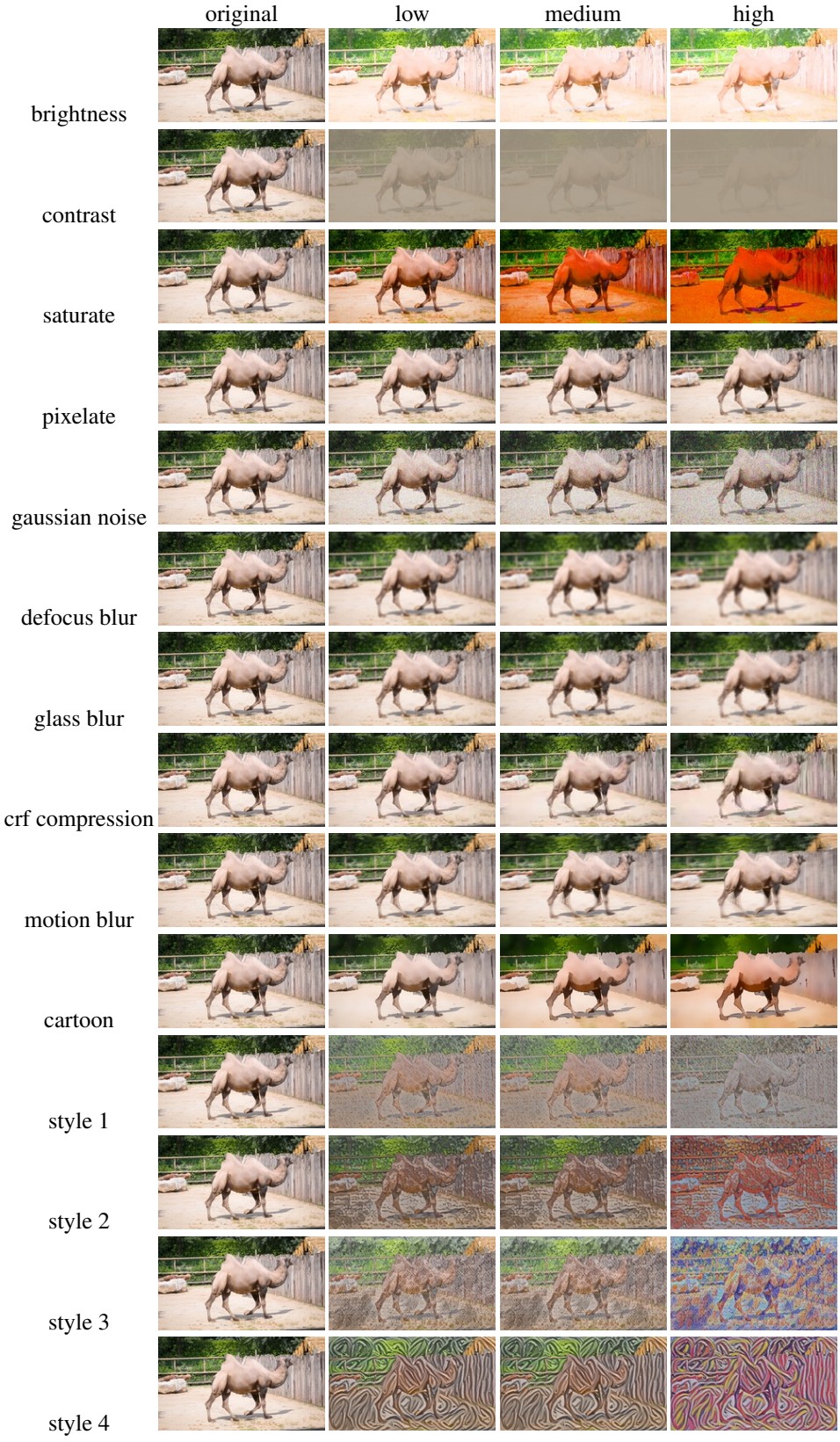

Figure 17: **Impact of the strength of the corruption** for the 14 types of corruption in DAVIS-C
.

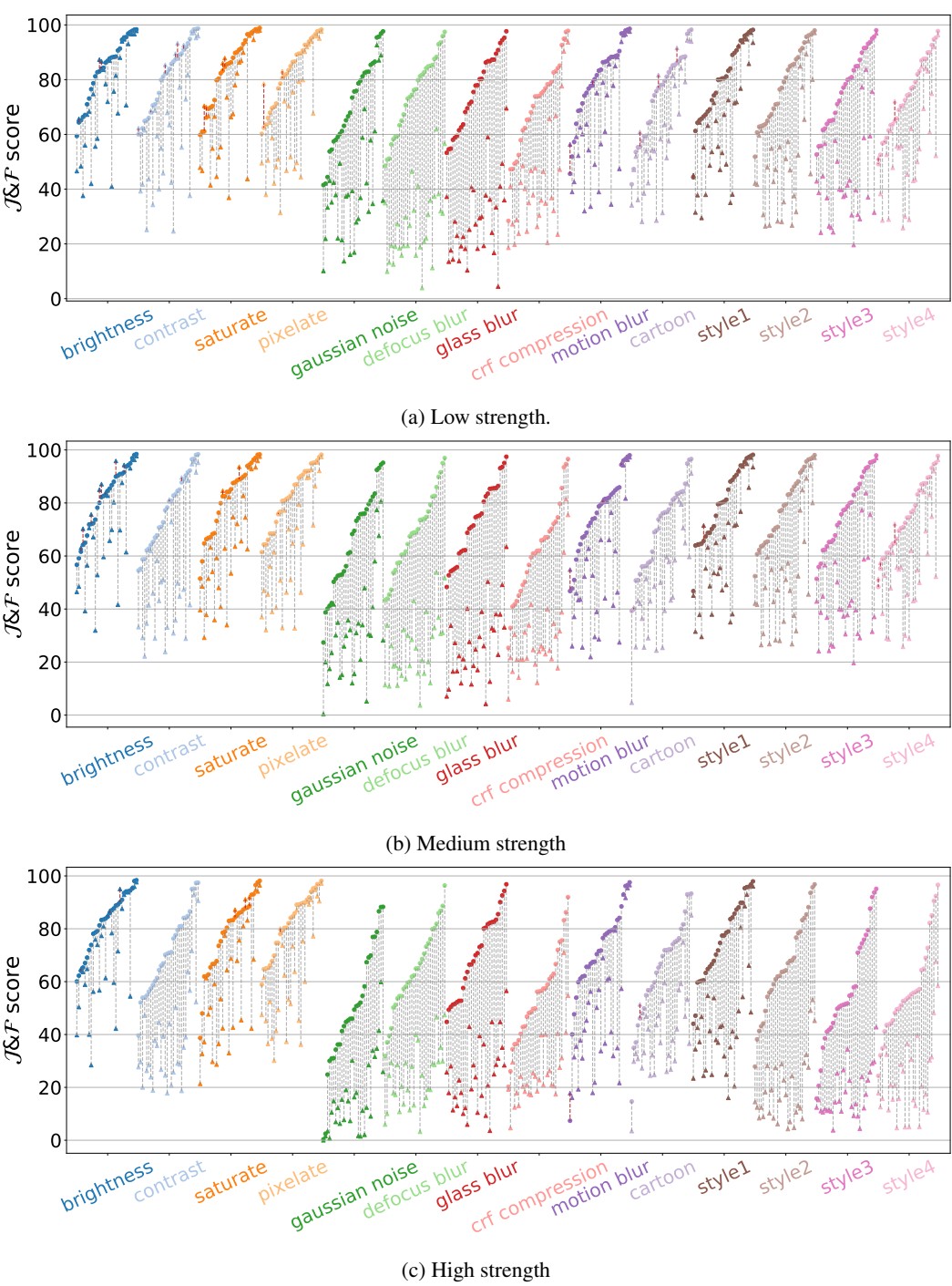

(a) Low strength.

(b) Medium strength

(c) High strength

Figure 18: **Test-time training on DAVIS-C for the STCN-BL30K model**. We plot the $\mathcal{J}\&\mathcal{F}$ score separately per video before (triangles) and after $tt$-MCC (circles) for the STCN-BL30K model on the 14 corruptions of the proposed DAVIS-C benchmark. We report results for the variants with the highest corruption strength. A red vertical line denotes that performance drops with test-time training.

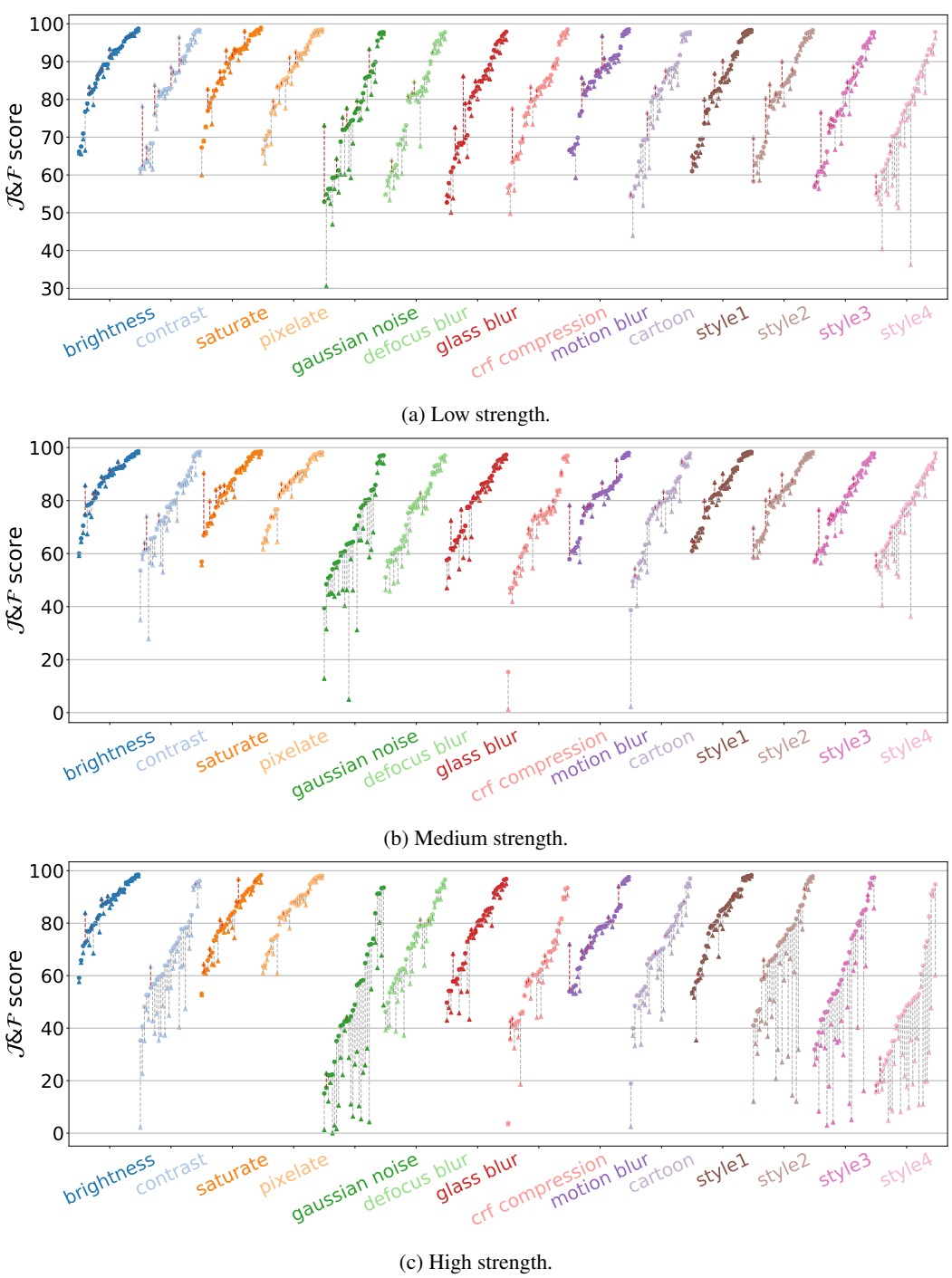

(a) Low strength.

(b) Medium strength.

(c) High strength.

Figure 19: **Test-time training on DAVIS-C for the STCN-DY model**. We plot the $\mathcal{J}\&\mathcal{F}$ score separately per video before (triangles) and after $tt$-MCC (circles) for the STCN-DY model on the 14 corruptions of the proposed DAVIS-C benchmark. We report results for the variants with the highest corruption strength. A red vertical line denotes that performance drops with test-time training.

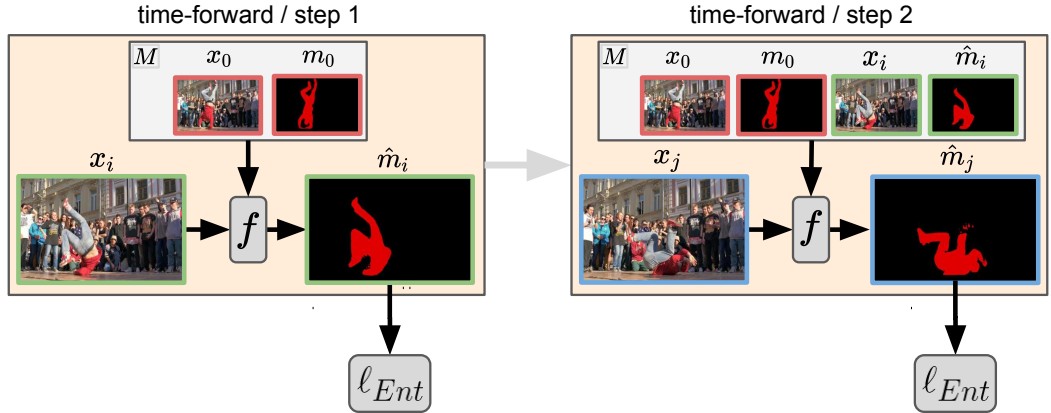

(a) **The $tt$-Ent loss** for a given frame triplet $\{x_0, x_i, x_j\}$. The frames $x_0, x_i$ and $x_j$ are shown with a red, green and blue border, respectively.

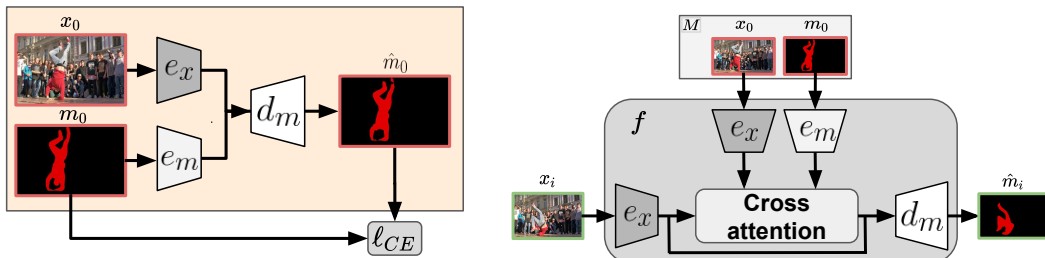

(b) **The $tt$-AE loss** for a given first frame $x_0$ and mask $m_0$. The functions $e_x$ and $e_m$ represent the frame and mask encoders. The function $d_m$ represents the mask decoder. They are components of function $f$ which is detailed on the right for STCN.

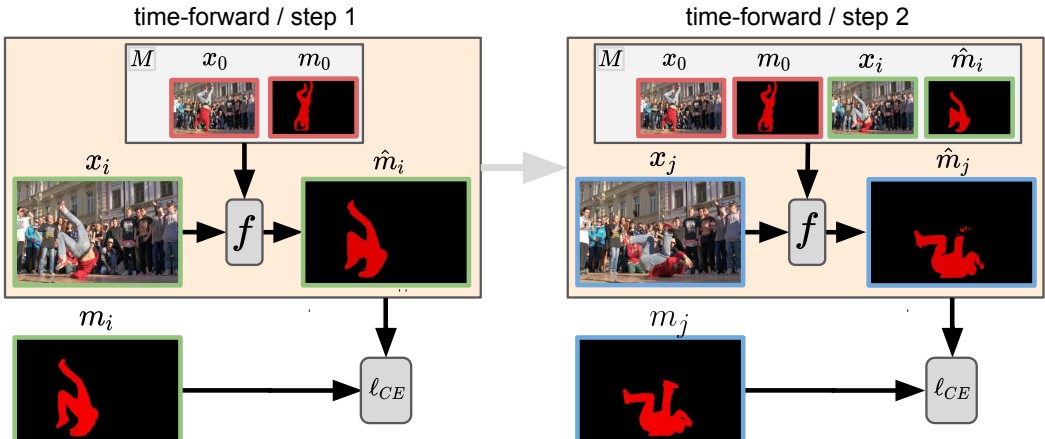

(c) **The regular cross-entropy loss** for a given frame triplet $\{x_0, x_i, x_j\}$ and mask pair $\{m_i, m_j\}$. The frames $x_0, x_i$ and $x_j$ are shown with a red, green and blue border, respectively.

Figure 20: **The $tt$-Ent, $tt$-AE, and the cross-entropy loss.** The function $f$ represents the overall prediction model. It takes as input the current test frame and a memory $M$ of predicted masks from the previous frames and outputs the predicted mask for the current frame.

