# OpenReview forum: "Test-time Training for Matching-based Video Object Segmentation"
_NeurIPS.cc/2023/Conference — NeurIPS 2023 poster_

### Official Review · Reviewer_jwwD · 2023-07-04

**Soundness:** 3 good
**Presentation:** 3 good
**Contribution:** 3 good
**Rating:** 7
**Confidence:** 5

**Summary:**

This paper investigates several test-time training methods for improving matching-based video object segmentation algorithms. It proposes and compares three types of test-time training methods and finds that the mask cycle loss works the best. The mask cycle loss forward/backward propagates a mask and requires cycle consistency. The authors find that test-time training significantly improves performance, especially in sim-to-real transfer and in videos with corruption.

**Strengths:**

- Adapting to harder cases during inference has been a challenge for video object segmentation algorithms. Test-time adaptation (i.e., online learning) can be useful but most of the current online learning methods are limited to fine-tuning on the frame-level without considering the temporal aspect. This means those methods are of limited aid when the video contains fast and complex motion of the target object. Differently, the proposed cycle loss takes the temporal information into account which is helpful.

- There are good analyses about how different models behave with different levels of data corruption.

- I like the inclusion of MOSE results in the supplementary material. It is a real-world challenging dataset that has much harder examples than the training set and current methods have difficulties generalizing to it. This shows that the proposed method works for real-world applications and the result is encouraging. I would recommend putting this in the main paper.


**Weaknesses:**

- There is no discussion on the time required for the fine-tuning stage and how it affects the overall running time during inference.

- The main results are presented with DAVIS-C which is synthetic. The results would carry more weight if done with the MOSE dataset instead.

- It would be more complete if the authors include a discussion on the use of data augmentation during the finetuning stage. Related: Lucid Data Dreaming for Video Object Segmentation, IJCV 2019.

- Missing related paper: Delving into the Cyclic Mechanism in Semi-supervised Video Object Segmentation, NeurIPS 2020


**Questions:**

- Table 3 in the supplementary material: the two bars with 38.9 do not have the same height. Is this a typo?

- Figure 3 – The mask predictions before test-time training look bad. Is that solely caused by the gaussian noise?


**Limitations:**

Test-time training is slow. Also, cycle loss can lead to degenerate output (e.g., identical mask at every time step) which might be why the authors only train for 100 steps and that some outputs become worse after finetuning (Figure 7).

---

> ### Author Rebuttal · Authors · 2023-08-09
>
> >  __Timings for the fine-tuning stage__
>
> We thank the reviewer for bringing attention to this. In the supplementary, we report that one iteration of tt-MCC takes ~814 ms for STCN (supp-line68). Timing estimates for a single video (on average and on DAVIS) are as follows: it takes roughly 3 seconds to run STCN per video, and 81, 10.5 and 8.5 seconds to run tt-MCC, tt-Ent, and tt-AE, respectively, for 100 iterations. Note however that: a) TTT is run per video, i.e. TTT overhead is amortized and becomes less for longer videos; b) decent improvements can be obtained with less than 100 TTT iterations, e.g. see Table A of the rebuttal PDF; c) the timings above are estimated with our basic, non-optimized TTT implementation. During the rebuttal, for example, we managed to obtain a speed-up of 20%, while further optimization is definitely possible.
>
> Although applying TTT consistently might seem infeasible due to time overheads, what our paper shows is that it is highly useful for cases of extreme test-time distribution shifts. One should therefore also consider the case where we are given a few out-of-distribution examples and want to quickly adapt our current model to improve its performance on them. In such cases, TTT is a simple and cost-effective solution, far preferable to re-training the base model (it takes around 12.5 hours on 2 A100 GPUs to train STCN).
>
> We thank the reviewer for their comment and we promise to clearly report and discuss the time needed for test time training in the paper.
>
> > __The main results are presented with DAVIS-C which is synthetic. The results would carry more weight if done with the MOSE dataset instead.__
>
> We agree that MOSE is a great new dataset (intricate occlusions, densely populated real-world settings). It is however not clear to us whether the reviewer suggests (a) to test on MOSE with corruptions, (b) to test on MOSE for the sim-to-real case, or (c) to test on MOSE for the case of no test-time distribution shift.
>
> With respect to (a), we agree that this would be valuable, but also very costly for experimentation since MOSE is many orders of magnitude larger than DAVIS and DAVIS-C contains 45 times more videos than DAVIS (for each original video we produce 15 corruptions over 3 levels). We can however consider this as a possible future update.
>
> With respect to (b), this is something already included in the supplementary material Figure 3; TTT is very valuable there as well. We can include this in the main paper if space allows.
>
> The case of (c) has not been the major focus of our paper, i.e. we focus on test-time distribution shifts. We did however run these experiments during rebuttal and the results are included in Table A of the rebuttal PDF with figures we submitted. There, we explore the use of a smaller number of iterations for TTT, which appears to be a better option for this setup and works for the case of testing on corruptions too. Let us further note here that it is common in the TTT literature to not report improvements for the case of no test-time distribution shift [18,35,39]; not decreasing the performance under no shift is typically considered an achievement ([39] states “Tent reaches the least error for most corruption types without increasing the error on the original data“).
>
> > __Discussion on the use of data augmentation during the finetuning stage. Related: Lucid Data Dreaming for Video Object Segmentation, IJCV 2019.__
>
> We thank the reviewer for a great comment and a related work we missed. Inspired by that work, we tested the impact of standard image augmentations (color+geometric) on our TTT method. Their use had little impact over all cases. In particular, we observed some insignificant performance increase in the sim-to-real case on DAVIS, and a similarly insignificant drop in performance for videos with corruptions from DAVIS-C. Note that the authors of that work claim that “Ideally training data should be as similar as possible to the test data, even subtle differences may affect quality” which fully aligns with the proposed MCC loss that makes use of the actual future video frames instead of trying to hallucinate them. We will add this experiment and discussion in the paper.
>
>
> > __Missing related paper: Delving into the Cyclic Mechanism in Semi-supervised Video Object Segmentation, NeurIPS 2020__
>
> Thank you. We will add the missing reference and discuss it. In short, the paper is using CC during training time, which allows to perform training while requiring masks only for the first video frame. This is similar to the way the cyclical loss is used in HODOR [1]. In contrast, we use CC for TTT, which allows us to perform TTT without any extra input than the one that is required by the task itself.
>
> > __Table 3 in the supplementary material: the two bars with 38.9 do not have the same height. Is this a typo?__
>
> Thank you for pointing this out. It is indeed a typo. The correct number for the MOSE train is _42.9_
>
> > __Figure 3 – The mask predictions before test-time training look bad. Is that solely caused by the gaussian noise?__
>
> This is correct. Gaussian noise in itself is a quite challenging corruption.

---

> > ### Comment · Reviewer_jwwD · 2023-08-10
> >
> > Thank you for the response. Do the authors observe degenerate output led by cycle loss with longer training (in limitation)?

---

> > > ### Author Response · Authors · 2023-08-12
> > >
> > > To better study the effect of longer training at test-time, we ran our method for up to 500 iterations on DAVIS for the sim-to-real case, extending Figure 7 from the main paper. In summary, we did not observe any significant further decrease in VOS performance nor any degenerate outputs after longer training. We will add this discussion in the paper.
> > >
> > > More specifically, the number of DAVIS videos for which VOS performance decreases remained the same from 50 TTT iterations up to approximately 400, i.e. the J&F score decreases in 7 out of 30 videos. Only one extra video showed a decrease in J&F score by 500 iterations. Visually observing segmentation results, we notice the same issues that we mention in our responses for Reviewers hK29 and 3wfH above: some background pixels of similar appearance to the object are wrongly segmented. It is also worth noting that increasing the iterations only leads to a minor overall decrease in VOS performance, i.e. J&F score drops from 81.2 to 81.0/80.4 for 200/500 iterations, respectively.

---

> > > > ### Comment · Reviewer_jwwD · 2023-08-12
> > > >
> > > > Thank you for the follow-up. The rebuttal addressed my concerns well and I have no further questions. I believe that the authors can include the running time information and the MOSE dataset result in the main paper which makes it more complete.
> > > >
> > > > I am raising the rating to Accept because I think this is a technically solid paper with a good impact in the sub-fields of robust video object segmentation/test-time training for videos with no major flaws. I have read the other reviews.

---

### Official Review · Reviewer_hK29 · 2023-07-05

**Soundness:** 3 good
**Presentation:** 3 good
**Contribution:** 3 good
**Rating:** 4
**Confidence:** 3

**Summary:**

This paper points out that Current state-of-the-art approaches use a memory of already processed frames and estimate the segmentation masks of follow-up frames through matching. Lacking any adaptation mechanism, such methods are prone to test-time distribution shifts. To address this, this paper  explores task-agnostic and VOS-specific test-time training strategies, including a mask cycle consistency-based variant tailored for matching-based VOS methods. The authors present a new dataset DAVIS-C and the proposed method is evaluated on the most common benchmarks, and demonstrate its effectiveness.

**Strengths:**

1. Experimental results show that the proposed approach (MCC) significantly boosts the performance.

2. Paper is very easy to read and understand.



**Weaknesses:**

1. Lack of technical contributions. Although mask cycle consistency usage at test time helps to boost the performance and enable more task-agnostic prediction, this seems to be a very similar technique that with minor modification (mask) from other works.

2. Just a question though. Why is XMeM not included in Fig 5?

3. Figure 6 and 7 are very hard to see and understand. I understand what these figures are trying to deliver, but to me, these figures definitely do not look so informative due to its seemingly unorganized graphs.

4. Ablations and limitations are missing. It would be much more informative to include failure cases as well.
What would happen if the method extends from triplets, and use more images for MCC?



**Questions:**

See weaknesses above

**Limitations:**

Limitations are missing

---

> ### Author Rebuttal · Authors · 2023-08-09
>
> > __Lack of technical contributions [...] this seems to be a very similar technique that with minor modification (mask) from other works__
>
> Although Cycle Consistency (CC) has been used in the past, our method is the first one to have such a loss tailored for the task of one-shot VOS, i.e. for a setting where a single _groundtruth_ mask is also given. In this fashion, the CC loss in our case is using privileged information and is supervised in contrast to many cases in prior work where it is self-supervised. This, together with the fact that our method is the first to apply this _at test-time_ we believe makes our tt-MCC method a solid technical contribution.
>
> Let us further note that, beside the technical contribution of the tt-MCC method, our paper is the first one to study the effect of extreme test-time distribution shifts like sim-to-real and frame-level or video-level corruptions for the task of one-shot VOS in a principled way. Additionally, our paper is the first to study test time training for VOS and includes benchmarking of a number of TTT baselines to offer valuable insights and a new benchmark for the community to build on and evaluate. Until now TTT has been mostly studied on image classification.
>
> > __Why is XMeM not included in Fig 5?__
>
> We did not include it because it cluttered the presentation, while providing little amount of additional information. The numbers are overall slightly higher, but the insights and trends remain. We will include this.
>
> > __Ablations and limitations are missing. It would be much more informative to include failure cases as well.__
>
> We thank the reviewer for a valid comment. We believe that the rebuttal PDF and video significantly expand our exploration and analysis on limitations and failure cases. After extensive quantitative analysis (some cases also included in the rebuttal video), the key failure modes can be summarized as: a) With TTT, the model might get more/overly confident about a dubious prediction and propagate it wrongly (e.g. see monkey clip in the video). b)  TTT might overfit the appearance module and wrongly add background pixels (e.g. see case of breakdancer, where a red shirt from the background confuses the model more than the STCN case).
>
> In Figure A of the rebuttal PDF, we provide some more in-depth analyses and a breakdown of the performance per video/object for the large MOSE-train dataset. In Figure A.a we evaluate performance gain with TTT vs the length of the video, as the reviewer suggested. We can observe that the video length has no significant impact on the performance. In Figure A.b, we present the performance gain vs the object size within the first frame and observe a slight correlation; for larger areas there is no negative gain.
>
> Inspired by the fact that the MOSE dataset contains examples where only a portion of the object is visible and annotated in the first frame due to occlusions, in Figure A.c we measure the impact of the size of the object in the first frame relative to the maximum size of the object within the video. A minor observation is that the gain is negatively impacted to a small extent when the object is less visible in the first frame compared to the subsequent frames. Unsurprisingly, the proposed tt-MCC overfits more to that partially annotated first frame, and usually segments the partial object in future frames instead of the full object. Note that extreme cases of severe occlusion in the first frame make such a VOS task ambiguous.
>
> Moreover, in Figure A.d of the rebuttal PDF we evaluate the impact of the object visibility within the video sequence, and confirm that tt-MCC is negatively impacted when the object is only briefly visible in the sequence. This could mean that by requesting long term temporal consistency, our CC loss imposes an inductive bias that the target object should be generally visible for a long period of time.
>
> Finally, Figure B of the rebuttal PDF explores temporal stability and presents a performance comparison over time on a per video basis, with and without TTT and also displays some failure cases.
>
> We will include all these additional analyses in the paper.
>
> > __What would happen if the method extends from triplets, and use more images for MCC?__
>
> We generally followed exactly the same design that each base method uses during training and adopted the use of triplets for STCN and octuplets for XMem. We briefly explore performance using longer sequences for STCN in the supplementary material (lines 60-62): Using quadruplets or quintuplets instead of triplets reaches 81.7 and 82.4, respectively, but with a corresponding increase of 33% and 64% in training time, i.e., in summary, having longer sequences does bring some minor improvements, but with the cost of additional TTT time. We can discuss this in the main paper if reviewers think we should.
>
> Let us also note that, instead of increasing the number of images, one can achieve longer temporal consistency by modulating the value of the jump step $s$: In the supplementary, we validate its impact (supp-line 55) to observe that neither very small nor very large values are the optimal choices:
> Varying the jump steps from the default value of 10 to 1, 2, 25, and 50 achieves 80.6, 81.0, 80.8, and 79.0, respectively. See paragraph “Impact of triplet sampling” in Section C of the supplementary for further exploration of the sequence sampling.

---

### Official Review · Reviewer_dPR9 · 2023-07-05

**Soundness:** 3 good
**Presentation:** 2 fair
**Contribution:** 2 fair
**Rating:** 5
**Confidence:** 4

**Summary:**

This paper focuses on test-time training for matching-based VOS methods. Three methods are presented, including entropy loss, auto-encoder loss and cycle consistency loss. The cycle-consistency loss is tailored for VOS and utilizes the first-frame mask for supervision. Two evaluation protocols are provided, the sim-to-real and DAVIS-C. Results show that the proposed tt-MCC method can help matching-based VOS methods to gain better performance under sim-real shifts and corrupted videos.

**Strengths:**

- The proposed method appears to be straightforward to implement and potentially effective in conjunction with matching-based VOS methods.

- The proposed DAVIS-C benchmark is a valuable contribution that can aid the community in advancing their research.


**Weaknesses:**

- **The proposed tt-MCC focuses solely on triplets and overlooks longer temporal consistency.** Methods that rely on short-term memory (STM) usually utilize multiple frames in their memory pools, allowing for aggregating long-term temporal information. However, tt-MCC only employs two frames for each training step, potentially limiting its exploitation of temporal consistency.

- **The concept of cycle consistency has been extensively explored and widely employed in various video-related fields.** In light of this, the proposed method is not groundbreaking. Moreover, a recent study [a] shares a similar idea and asserts that their approach outperforms cycle consistency (as shown in Table 7). The authors may want to highlight the disparities and advantages of tt-MCC compared to the method proposed in [a].

- **The results for XMem-YD or STCN-YD + tt-MCC under no-corruption conditions on large-scale benchmarks like YouTube-VOS and MOSE are missing.** Additionally, synthetic datasets are not as commonly used as static images for pretraining. It would be beneficial to examine whether tt-MCC aids matching-based VOS methods after static pretraining.

[a] Boosting Video Object Segmentation via Space-time Correspondence Learning, CVPR2023

**Questions:**

See Weaknesses.

**Limitations:**

The method does not include a mechanism for implementing early stopping and determining the optimal iteration to stop training.

---

> ### Author Rebuttal · Authors · 2023-08-09
>
> > __The proposed method focuses solely on triplets and overlooks longer temporal consistency.__
>
> We generally followed exactly the same design that each base method uses during training and adopted the use of triplets for STCN and octuplets for XMem. We briefly explore performance using longer sequences for STCN in the supplementary material (lines 60-62): Using quadruplets or quintuplets instead of triplets reaches 81.7 and 82.4, respectively, but with a corresponding increase of 33% and 64% in training time, i.e., in summary, having longer sequences does bring some minor improvements, but with the cost of additional TTT time. We can discuss this in the main paper if reviewers think we should.
>
> Let us however also note that one can achieve longer temporal consistency even using triplets, by modulating the value of the jump steps: In the supplementary, we validate its impact (supp-line 55) to observe that neither very small nor very large values are the optimal choices:
> Varying the jump steps from the default value of 10 to 1, 2, 25, and 50 achieves 80.6, 81.0, 80.8, and 79.0, respectively. See paragraph “Impact of triplet sampling” in Section C of the supplementary for further exploration of the sequence sampling.
>
> > __Cycle consistency has been extensively explored - relation to [a] “Boosting Video Object Segmentation via Space-time Correspondence Learning” (CVPR 2023)__
>
> We thank the reviewer for pointing out an interesting related work that is concurrent to ours. The key differences between [a] and our approach are: a) unlike our paper, [a] does not study the case of test-time adaptation of a model for VOS,  and b) we use the cycle consistency on the _mask_ labels, not on the pixels or regions. [a] builds on other related papers like [16] where space-time consistency is imposed on the rgb pixels and further extends it to regions, but does not use the mask as a label.
>
> > __The results for XMem-YD or STCN-YD + tt-MCC under no-corruption conditions on large-scale benchmarks like YouTube-VOS and MOSE are missing__
>
> We report the missing results in Table A of the rebuttal PDF, for a different number of test time training iterations. In the main paper, our core focus is on the case of test-time distribution shifts,  and we only report results for 100 iterations which we found to be best for all cases with distribution shifts. For the case where there is no distribution shift, however, we see from Table A that, unsurprisingly, using less iterations could be beneficial, especially for the larger scale YTVOS and MOSE datasets. Overall we see that TTT does not hurt the state of the art methods in this case, but instead gives us a way of either preserving or improving SoTA performance in all cases of test-time videos, with or without distribution shift, something important in practice. We thank the reviewer and we will add these results.
>
> Note that a large number of existing papers for TTT in classification ([18][35][39]) report no or insignificant improvements for the case of no test-time distribution shifts. Actually, not decreasing the performance under no shift is typically considered an achievement ([39] states “Tent reaches the least error for most corruption types without increasing the error on the original data“)
>
>
> > __examine whether tt-MCC aids matching-based VOS methods after static pretraining__
>
> We thank the reviewer for a valuable additional experiment. We explored this during rebuttal and report results in the right part of Table A in the rebuttal pdf, for STCN and XMem on 4 datasets. We conclude that tt-MCC provides significant improvements in this setup too. We will add these experiments.

---

> > ### Comment · Reviewer_dPR9 · 2023-08-16
> >
> > I sincerely thank the authors for providing a detailed response to my concerns. Given that the concept of Cycle Consistency has already been extensively studied in various tasks, including semi-supervised VOS, the originality of the approach in this paper is somewhat diminished. However, the experimental results presented in both the paper and the rebuttal adequately demonstrate the effectiveness of the proposed training strategy on STM-based methods [5,7]. Therefore, I am inclined to stick with the initial borderline accept score.
> >
> > Questions raised during rebuttal:
> > - It appears that the author has only validated the effectiveness of their proposed method on STM-based VOS methods [5,7] without conducting experiments on other types of VOS methods, such as recent AOT-based methods [46,48]. Is it possible that the proposed method is only applicable to STM-based methods?

---

> > > ### Author Response · Authors · 2023-08-17
> > > **Official Comment by Authors**
> > >
> > > This is a good question. The main variant, tt-MCC, is well-suited for all matching-based methods that adhere to the function $\hat{m}_j = f(x_j, M)$, where $M$ represents the memory of previous frame predictions (line 120 of our paper). Upon revisiting the AOT method, we conclude that its characteristics align with the requirements of our tt-MCC formulation. However, we recognize that the integration of tt-MCC into the AOT implementation is a task that transcends the scope of the rebuttal period. We view this potential integration as an interesting task for future research or potential extensions of our own work.
> > >
> > > In contrast, tt-Ent, another proposed variant, possesses an even broader applicability beyond the realm of matching-based methods. Regarding tt-AE, as stated in our paper, it is tailored to the STCN architecture and does not possess a direct out-of-the-box applicability to AOT. However, we acknowledge the possibility of a modified variant that could be designed to align with the AOT architecture. While we have not yet conducted an exhaustive analysis of all the aspects involved, we believe that a carefully tailored adaptation is plausible.
> > >
> > > Moreover, please refer to our earlier response to reviewer iPsr which states that “our formulation is generic and applicable to a wide range of matching-based VOS methods, including ones that require only a single forward pass for multiple objects, e.g. AOT [48].“.

---

> > > > ### Comment · Reviewer_jwwD · 2023-08-17
> > > >
> > > > I agree with reviewer dPR9 that an extension to AOT (and see if it works) can be interesting. Given that they are quite different architectures and are implemented differently, there will be significant technical challenges in doing it within a short time. I think this extension is not necessary to prove the effectiveness of the proposed method (though it would strengthen the conclusion) and is out-of-scope for the purpose of this paper submission. The high-level justification/motivation that works for STCN should also work for AOT, and STCN has the benefit of being one of the simplest baselines out there that can be analyzed easily.

---

> > > > > ### Comment · Reviewer_dPR9 · 2023-08-21
> > > > >
> > > > > Thanks for the follow-up. I agree with R-jwwD that an extension to AOT is not simple work. I was seeking more contributions that the authors may provide, and thus curious if they had tried to validate their method on different architectures. Although the authors have not done this, related discussions will still benefit the main paper.

---

> > > > > > ### Author Response · Authors · 2023-08-21
> > > > > > **Thank you for the discussion**
> > > > > >
> > > > > > We thank both reviewers for the constructive discussion. We agree that discussing other architectures will benefit the main paper and we promise to add such a discussion to it. Also, as we mention above, we are still further considering running experiments over AOT for the final version.

---

### Official Review · Reviewer_iPsr · 2023-07-06

**Soundness:** 3 good
**Presentation:** 4 excellent
**Contribution:** 3 good
**Rating:** 6
**Confidence:** 5

**Summary:**

This paper revisits test-time training in video object segmentation (VOS) and introduces three losses (entropy loss, mask auto-encoder loss, and mask cycle consistency loss) that significantly enhance top-performing methods, particularly under extreme distribution shifts between training and testing sets. Additionally, it introduces DAVIS-C, a variant of the DAVIS test set with extreme distribution shifts, demonstrating great performance on this challenging dataset.

**Strengths:**

1. The proposed method enhances test-time training for matching-based video object segmentation (VOS) and achieves significant performance improvements for state-of-the-art methods in scenarios with extreme distribution shifts (e.g., synthetic to real data, stylization, and video corruptions).
2. The paper introduces the DAVIS-C dataset, specifically designed to evaluate model performance under extreme distribution shifts during test time.
3. The proposed method achieves a 70%-80% recovery in performance gain by training solely on synthetic data, without the need for video annotations during offline training.

**Weaknesses:**

Please see my comments in below Questions section.

**Questions:**

1. It would be beneficial to clarify how the proposed model can be extended to handle multiple objects. Specifically, it is important to address whether multiple object segmentation can be achieved in a single pass during the inference step or if multiple forwards are required.

2. I kindly request that the authors report the finetuning time alongside Figure 7. It would be more informative to have the exact time instead of just the number of training iterations. This information is valuable since one of the main drawbacks of test-time training methods is the potentially long finetuning time during test runs. Additionally, please clarify if the reported average inference time with TTT per frame includes the finetuning time, as mentioned in lines 227-228.

3. Please kindly provide information regarding the memory requirements of the proposed method. Since it stores both mask and frame representations, it would be helpful to know if there is a threshold or mechanism to manage memory usage and if any data is dropped when reaching that threshold.

4. In lines 129-131, please provide clarification regarding the similarity matching metric utilized to identify similar items.

5. In line 160-166, the last frame prediction is dependent on the second frame prediction, I am wondering if this type of "dependency" is necessary? For example, is it possible to predict both the last mask and the second mask based solely on the first mask?

---

> ### Author Rebuttal · Authors · 2023-08-09
>
> > __Clarify how the proposed model can be extended to handle multiple objects__
>
> The proposed test-time training (TTT) method is a way of performing test-time adaptation over a given base method (STCN/XMem in our paper). It does not change the way the base method does inference, and for STCN and XMem, for example, multiple objects are handled sequentially. Nevertheless, our formulation is generic and applicable to a wide range of matching-based VOS methods, including ones that require only a single forward pass for multiple objects, e.g. AOT [48]. In such a case, multiple objects would be handled via a single forward pass for TTT as well.
>
>
> > __Finetuning/TTT time__
>
> We thank the reviewer for bringing attention to this. The average inference time reported in lines 227-228 of the main paper is _after_ TTT. We tried to clarify timings better in the supplementary, i.e. see lines 66-71; we apologize for any confusion. In the supplementary, we report that one iteration of tt-MCC takes ~814 ms for STCN (supp-line68). With this information, one can translate the x-axis of Figure 7 from iterations to seconds; we will update this in the final version.
>
> Timing estimates for a single video (on average, on DAVIS) are as follows: it takes roughly 3 seconds to run STCN per video, and 81, 10.5 and 8.5 seconds to run tt-MCC, tt-Ent, and tt-AE, respectively, for 100 iterations. Note however that: a) TTT is run per video, i.e. TTT overhead is amortized and becomes less  for longer videos; b) decent improvements can be obtained with less TTT iterations, e.g. see Table A of the rebuttal PDF; c) the timings above are estimated with our basic, non-optimized TTT implementation. During the rebuttal, for example, we managed to obtain a speed-up of 20%, while further optimization is definitely possible.
>
> Although applying TTT consistently might seem infeasible due to time overheads, what our paper shows is that it is highly useful for cases of extreme test-time distribution shifts. One should therefore also consider the case where we are given a few out-of-distribution examples and want to quickly adapt our current model to improve its performance on them. In such cases, TTT is a simple and cost-effective solution, far preferable to re-training the base model (it takes around 12.5 hours on 2 A100 GPUs to train STCN).
>
> Overall, we fully agree with the reviewer that long fine tuning times are a key drawback for test-time training methods in general, and one that the community does not even touch: Most TTT papers do not have any discussion on timings, although it is something important to report and discuss. We thank the reviewer for their comment, and we promise to clearly report and discuss the time needed for test time training in the paper.
>
> >__Memory requirements__
>
> During inference, STCN requires approx. 8GB of GPU RAM. Test-time training on top of STCN with tt-AE, tt-Ent and tt-MCC requires 8,16 and 23.5GB, respectively, i.e. it can still fit in a modest GPU. Note that the calculations for TTT are computed when using a batch size of 4 (our default). Memory requirements can be further reduced by using a smaller batch size, if needed, without significant change in performance.
>
> We thank the reviewer for such comments on timings and memory; we will make sure to clearly report and discuss the added resources needed for test time training.
>
> > __Clarification regarding the similarity matching metric__
>
> The matching part is inherited from the base method. The technical details are exactly the same as in STCN and are described in the original paper. In particular, the full similarity matrix is formed between the test frame items and the memory items but only the top-k values per row (rows correspond to the test frame) are maintained, the rest are set to 0. By the term item we refer to each D_e-dimensional vector, with each frame having WxH of them.
>
> > __Is it possible to predict both the last mask and the second mask based solely on the first mask?__
>
> Yes, this is technically possible. We did not experiment with that, as we adopt the design choices of STCN. We expect its performance to not vary significantly.

---

> > ### Comment · Reviewer_iPsr · 2023-08-20
> >
> > After reviewing the rebuttal and considering the comments from the other reviewers, I have increased my rating to Weak Accept. I value the author's response regarding test time and memory concerns. Kindly ensure that this aspect is reported and discussed clearly in the revised version, as promised. This is of great importance for a method of this nature.

---

> > > ### Author Response · Authors · 2023-08-21
> > > **Thank you for raising your rating**
> > >
> > > We thank the reviewer for acknowledging our rebuttal and for increasing their rating. We will include a clear discussion on test time and memory consumption in the main paper as promised.

---

### Official Review · Reviewer_3wfH · 2023-07-06

**Soundness:** 3 good
**Presentation:** 2 fair
**Contribution:** 2 fair
**Rating:** 4
**Confidence:** 4

**Summary:**

The paper introduces an adaptation/test-time training algorithm for matching-based video object segmentation, specifically addressing the challenges posed by out-of-distribution videos(corruptions, stylization, and sim-to-real transfer). The proposed framework incorporates test-time training with various losses (entropy, mask consistency, etc.) The experimental evaluation includes benchmark datasets like DAVIS and youtube-VOS, where the authors demonstrate performance improvements for out-of-distribution scenarios.

**Strengths:**

- The proposed direction is highly interesting and carries significant practical implications.

- The results obtained are quite promising. Notably, there are substantial improvements observed across multiple datasets, particularly in the case of corrupted DAVIS. These outcomes highlight the effectiveness and value of test-time training in this context.

**Weaknesses:**

The technical aspects of the paper may appear somewhat limited. While concepts such as test-time training and loss are well-known techniques, the actual technical contribution seems marginal. It would be beneficial to explicitly highlight the distinctions between this work and previous approaches.

The validation process also seems somewhat limited, which undermines the full convincing power of the proposed method, in particular:
- The absence of qualitative results in video formats makes it difficult to assess the practical impact and temporal stability of the improvements achieved.
- The resemblance of DAVIS-C to real-world noise is uncertain, as the augmentation techniques seem quite artificial. Additionally, it may be difficult to reproduce the study on DAVIS-C, and the argumentation process is not fully clear.
- Providing more in-depth analysis on intermediate/breakdown results would be valuable. For instance, exploring the performance relative to video length, examining corner cases where the method fails, and identifying scenarios where the proposed approach is most beneficial. Currently it's hard to gain further insights from the results.

**Questions:**

Please discuss the questions raised above.

**Limitations:**

It has been discussed however not fully clear what the failure mode of the approach would be, might be helpful to conduct more analysis on this.

---

> ### Author Rebuttal · Authors · 2023-08-09
>
> > __Distinctions between this work and previous approaches__
>
> Although Cycle Consistency (CC) has been used in the past, our method is the first on to have such a loss tailored for the task of one-shot VOS: Unlike methods like “Space-time correspondence as a contrastive random walk” [16] or the concurrent work “Boosting Video Object Segmentation via Space-time Correspondence Learning” from CVPR 23, we do not apply the cycle-consistency loss on RGB pixel values, but instead on the values of the mask in the first frame.  This means that, unlike space-time correspondence CC losses, ours is not self-supervised. Together with the fact that our method is the first to apply this _at test-time_ we believe makes our tt-MCC method a solid technical contribution.
>
> Unlike other methods like HODOR [1] that might employ a cycle consistency loss during training, our method tailors this to test time training and uses the GT frame that is provided at test-time. Concurrent work [49] also studies test time training (TTT) in the video domain but for a classification task; we instead fully tailor the TTT on the VOS task and use temporal _mask consistency_ as our loss. We hope that this answer covers the differences between the proposed and the closest related works, if the reviewer has another in mind, please let us know.
>
> Beside the technical contribution of the tt-MCC method, we want to further note here that our paper is the first one to study the effect of extreme test-time distribution shifts like sim-to-real and frame-level corruptions for the task of one-shot VOS in a principled way. Additionally, our paper is the first to study test time training for VOS and includes benchmarking of a number of TTT baselines to offer valuable insights and a new benchmark for the community to build on and evaluate. Until now TTT has been mostly studied on image classification.
>
> > __Absence of qualitative results in video formats makes it difficult to assess the practical impact and temporal stability__
>
> This is a great point. We are providing multiple qualitative results in the rebuttal video shared with the AC. It enables comparing results with and without TTT visually for a number of corner cases. From the rebuttal video and after extensive quantitative analysis, the key failure modes can be summarized as: a) With TTT, the model might get more/overly confident about a dubious prediction and propagate it wrongly (e.g. see monkey clip in the video). b)  TTT might overfit the appearance module and wrongly add background pixels (e.g. see case of breakdancer, where a red shirt from the background confuses the model more than the STCN case).
>
> Regarding temporal stability, we include in Figure B of the rebuttal PDF a performance comparison over time on a per video basis, with and without TTT. It demonstrates enduring performance gains. Notably, it also shows that the performance with TTT is more stable compared to that without TTT.
>
>
> > __The resemblance of DAVIS-C to real-world noise is uncertain__
>
> Thank you for a valuable comment. We agree that these transformations might not perfectly represent the outcome of video recording in the real-world. Nevertheless, we believe that several of the transformations do constitute common real-world video edits (contrast, brightness, crf compression, cartoonization, stylization).
>
> We follow the footsteps of ImageNet-C which has been a valuable benchmark to study image classification under distribution shift. We believe that this benchmark can lead to valuable insights, similar to what ImageNet-C achieved for image understanding.
>
>
> > __it may be difficult to reproduce the study on DAVIS-C__
>
> We are committed to releasing the DAVIS-C dataset, the code to regenerate the dataset, and the code for this study, which will make it possible to not only reproduce this study, but also make it easy to evaluate any future method on the same benchmark in a fair way.
>
>
> > __More in-depth analysis on intermediate/breakdown results would be valuable__
>
> Great suggestion. In Figure A of the rebuttal PDF, we provide some more in-depth analyses and a breakdown of the performance per video/object for the large MOSE-train dataset. In Figure A.a we evaluate performance gain with TTT vs the length of the video, as the reviewer suggested. We can observe that the video length has no significant impact on the performance. In Figure A.b, we present the performance gain vs the object size within the first frame and observe a slight correlation; for larger area there is no negative gain.
>
> Inspired by the fact that the MOSE dataset contains examples where only a portion of the object is visible and annotated in the first frame due to occlusions, in Figure A.c we measure the impact of the size of the object in the first frame relative to the maximum size of the object within the video.  A minor observation is that the gain is negatively impacted to a small extent when the object is less visible in the first frame compared to the subsequent frames. Unsurprisingly, the proposed tt-MCC overfits more to that partially annotated first frame, and usually segments the partial object in future frames instead of the full object. Note that extreme cases of severe occlusion in the first frame make such a VOS task ambiguous.
>
> Finally, we evaluate the impact of the duration of the object visibility within the video sequence in Figure A.d and confirm that tt-MCC is negatively impacted when the object is only briefly visible in the sequence. This could mean that by requesting long term temporal consistency, our CC loss imposes an inductive bias that the target object should be generally visible for a long period of time.

---

> > ### Comment · Reviewer_3wfH · 2023-08-20
> >
> > I would like to thank authors' for the feedback. Especially, I appreciate the discussion to contrast previous approaches, and additional analysis in figure A
> >
> > After reading the rebuttal as well as other reviews, I still have some concerns regarding the validation, and effectiveness, therefore I have adjusted my rating and am inclined to vote for rejection.

---

> > > ### Author Response · Authors · 2023-08-20
> > > **Thanks for the rating increase - happy to provide any further clarifications possible**
> > >
> > > We thank the reviewer for the response and for the rating increase. We would further appreciate it if the reviewer can share details regarding the remaining concerns so that we can provide additional clarifications. The reviewer's initial concerns regarding validation and effectiveness, as well as a brief summary of our responses to each one of them follows.
> > >
> > > * __“qualitative results and temporal stability”__: we provided an anonymized video with a large number of qualitative results for different success and corner cases, as well as Figure B of the rebuttal PDF showing that TTT is at least as stable as the base approach.
> > > * __"type of augmentations and reproducibility of DAVIS-C"__: we motivated the realistic aspect of the augmentations and guaranteed the reproducibility of the dataset and approach by committing to publicly share both implementations and the dataset itself. Additionally, please note that DAVIS-C is only one part of the evaluation study. The sim-to-real is the second part which was now strengthened with the case of training on static images as requested by Reviewer dPR9, showing that our methods are _effective_ also in this case.
> > > * __”more in-depth analysis”__: Figures A and B in the rebuttal PDF break down performance across different aspects (video length, object area, visibility duration) and report results across time, respectively. We are glad that the reviewer stated that they appreciate it.

---

### Author Rebuttal · Authors · 2023-08-09

We would like to thank all five reviewers for insightful and constructive reviews.

We are pleased that the feedback is __overall positive__. Four out of five reviewers highlighted the significant performance improvements of our proposed method under test-time distribution shifts. We are happy that they also found our paper “highly interesting” and with “significant practical implications” (__R-3wfH__), as well as our paper “easy to read and understand” (__R-hK29__). Reviewers further appreciated the introduction of the DAVIS-C benchmark (__R-iPsr__), calling it a “valuable contribution” for the community (__R-dPR9__). They highlighted our analyses over different levels of frame-level corruptions (__R-jwwD__) and the large improvements our method can achieve in such cases (__R-3wfH__).

We are responding to each reviewer individually below, addressing all of their questions and comments.

We will additionally send a video to the AC (referred to as __rebuttal video__ in the responses below) using the provided functionality, that summarizes the qualitative performance before and after our TTT for success, failure and other corner cases.

We further attach to this response a one-page rebuttal PDF document with additional results and plots (referred to as __rebuttal PDF__ in the responses below) that help us respond to reviewer comments. It contains:
* Table A with results for STCN and XMem on four datasets, for the cases of training from static images and real videos, as well as for a reduced number of iterations of test-time training.
* Figure A with four different statistical analyses for the sim-to-real case on the large MOSE dataset, demonstrating the performance of the proposed approach versus video length, object size as well as other key elements that help us demonstrate better where our method helps and where it fails.
* Figure B with plots that compare performance over time for a few videos, that justify the improved temporal stability of the proposed method and also demonstrate success and failure cases.

We hope that our responses below, together with all the additional qualitative and quantitative results and analysis, address the reviewers’ comments in a satisfactory way. We are more than happy to engage in further discussion and clarify any points that might still remain unclear.

---

### Decision · Program_Chairs · 2023-09-21

**Decision:**

Accept (poster)

**Comment:**

The paper investigates test-time training in video object segmentation. The experiments show performance improvement particularly in scenarios of sim-to-real transfer and corrupted videos.

This paper underwent a thorough review process by five reviewers. It received mixed ratings of 7, 6, 5, 4, and 4, placing it in a borderline category. The AC read the paper, the reviews, and the rebuttals. The rebuttals addressed concerns raised by the reviewers about speed, technical contributions, and discussions with prior works. Moreover, to validate the effectiveness the authors provided more qualitative results and a thorough ablation on four datasets including DAVIS, DAVIS-C, YTVOS and MOSE. The authors also promised to release the proposed DAVIS-C dataset and the code for reproduction of the research to address the reviewer's concern about reproducing the results for validation.

The AC agrees with the majority of the reviewers that the paper is technically sound and recommends it for acceptance. The authors should add all the additional results including the speed, memory consumption and the ablations in the final version.